# p50-associated COX-2 extragenic RNA (PACER) activates *COX-2* gene expression by occluding repressive NF-κB complexes

Michal Krawczyk, Beverly M Emerson*

Regulatory Biology Laboratory, Salk Institute for Biological Studies, La Jolla, United States

**Abstract** Deregulated expression of COX-2 has been causally linked to development, progression, and outcome of several types of human cancer. We describe a novel fundamental level of transcriptional control of *COX-2* expression. Using primary human mammary epithelial cells and monocyte/macrophage cell lines, we show that the chromatin boundary/insulator factor CTCF establishes an open chromatin domain and induces expression of a long non-coding RNA within the upstream promoter region of COX-2. Upon induction of *COX-2* expression, the lncRNA associates with p50, a repressive subunit of NF-κB, and occludes it from the COX-2 promoter, potentially facilitating interaction with activation-competent NF-κB p65/p50 dimers. This enables recruitment of the p300 histone acetyltransferase, a domain-wide increase in histone acetylation and assembly of RNA Polymerase II initiation complexes. Our findings reveal an unexpected mechanism of gene control by lncRNA-mediated repressor occlusion and identify the COX-2-lncRNA, PACER, as a new potential target for COX-2-modulation in inflammation and cancer.

**\*For correspondence:**
emerson@salk.edu

**Competing interests:** The authors declare that no competing interests exist.

**Reviewing editor**: Joaquin M Espinosa, HHMI/University of Colorado at Boulder, United States

## Introduction

Long non-coding RNAs (lncRNAs) are a highly heterogeneous class of RNA molecules which are >200nt long and, apart from the lack of protein coding potential, share many characteristics with mRNAs in that they are transcribed by RNA polymerase II (RNAP II) and can be capped, spliced, and polyadenylated (*Guttman et al., 2009*). LncRNAs can be located in intergenic DNA, within introns, or overlapping other genes in an antisense orientation and their expression is often tightly regulated and/or restricted to specific tissues or stages of development (*Guttman et al., 2009*; *Dinger et al., 2008*; *Mercer et al., 2008*; *Tsai et al., 2011*). Contrary to their smaller counterparts (tRNA, siRNA, miRNA etc), whose functions have been firmly established, the roles lncRNAs play in many biological phenomena are much less well understood. While examples exist of lncRNA involvement in such diverse processes as the cell cycle, cell migration and survival, metabolism, organization of subcellular compartments, among others, the predominant function of many lncRNAs appears to be regulation of gene expression (*Mercer et al., 2009*; *Wang and Chang, 2011*; *Ponting et al., 2009*; *Wilusz et al., 2009*).

LncRNAs can utilize a variety of mechanisms to regulate gene activity, where perhaps the most widespread involves interacting and recruiting repressive histone methyltransferase or demethylase complexes such as polycomb repressive complexes 1 and 2 (PRC1, PRC2), LSD1/CoREST/REST or G9a to target genes (*Wutz et al., 2002*; *Nagano et al., 2008*; *Khalil et al., 2009*; *Zhao et al., 2010*; *Tsai et al., 2010*; *Yap et al., 2010*). Conversely, another lncRNA was shown to interact with an activating HMT, the trithorax complex WDR5/MLL1 (*Wang et al., 2011*). In addition, a growing number of studies implicate lncRNAs in transcriptional interference, splicing, miRNA squelching, and direct interaction with hormone receptors, transcription factors, and other RNA-binding proteins (*Mercer et al., 2009*; *Wang and Chang, 2011*; *Ponting et al., 2009*; *Wilusz et al., 2009*; *Carpenter et al., 2013*).

**eLife digest** To produce a protein a cell must first transcribe the DNA in a gene to make a messenger RNA molecule, which is then translated to make the protein. However, cells also produce other types of RNA molecules which do not become proteins. MicroRNAs, for example, regulate the expression of genes as proteins, while the role of other RNA molecules called long non-coding RNAs (lncRNAs) is not well understood.

Now Krawczyk and Emerson have found an lncRNA that controls a gene called *COX-2* that is often implicated in breast, colon, prostate, and lung cancer. This RNA molecule, which is called PACER, originates near the start of the *COX-2* gene, but it cannot be messenger RNA because it does not contain the instructions to make the COX-2 protein, and it is too long to be a microRNA. Further experiments showed that the newly discovered lncRNA activates the expression of the *COX-2* gene.

Krawczyk and Emerson found that PACER attracts enzymes that spotlight genes that need to be activated, thus increasing the transcription of these genes to make messenger RNA. Genes can also be switched on and off by various molecules binding to nearby DNA, and PACER encourages the activation of *COX-2* by keeping away the molecules that might switch it off.

In addition to shedding new light on the role of lncRNAs, these results suggest that PACER could be a suitable therapeutic target in cancers that involve the *COX-2* gene.

Given that lncRNAs have gained increasing appreciation as key regulators of gene expression, it is not surprising that they are frequently deregulated during tumorigenesis (*Wapinski and Chang, 2011*; *Gibb et al., 2011*). Examples include H19, whose upregulation is detected in a variety of cancers (*Fellig et al., 2005*; *Barsyte-Lovejoy et al., 2006*; *Hibi et al., 1996*; *Adriaenssens et al., 1998*; *Kondo et al., 1995*); HOTAIR, whose overexpression is observed in breast, prostate, and other cancers and whose loss inhibits cancer invasiveness (*Gupta et al., 2010*); GAS5, an apoptosis regulator that is downregulated in breast cancers (*Mourtada-Maarabouni et al., 2009*); and ANRIL, whose expression is affected by SNPs that correlate with several neoplasias and other diseases (*Cunnington et al., 2010*). Despite the apparent functional consequences of lncRNA deregulation, very little is currently known about how expression of lncRNAs is normally modulated and what the basis is for its deregulation in cancer and other human diseases.

The CCCTC-binding factor CTCF is a highly conserved 11-zinc finger DNA binding protein. Its primary function is believed to involve formation of chromatin boundaries/insulators through chromosomal looping. However, CTCF has been implicated in a variety of other processes, including transcription, enhancer blocking, insulation, splicing, nucleolar maintenance, DNA methylation, and imprinting (*Merkenschlager and Odom, 2013*; *Phillips and Corces, 2009*; *Ohlsson et al., 2010*). Genomic profiling has shown that CTCF binds to tens of thousands of genomic sites located at gene promoters, within coding sequences and in intergenic regions (*Kim et al., 2007*; *Xie et al., 2007*; *Barski et al., 2007*). CTCF sites are enriched for histone variants H2A.Z and H3.3 and correlate with increased chromatin accessibility, nucleosomal depletion, presence of histone variants H2A.Z and H3.3 and histone H3K4 methylation (*Merkenschlager and Odom, 2013*).

Cyclooxygenase 2 (COX-2, also called prostaglandin-endoperoxide synthase 2, PTGS2) is one of two COX isomers that are key enzymes in prostaglandin (PG) biosynthesis. COX enzymes catalyze conversion of arachidonic acid to PGH2 that acts as a substrate for a number of eicosanoid derivatives, such as PGE2, PGI2 and thromboxane A2, which are important mediators of many biological processes including inflammation, fever, pain, gastric and kidney function (*Schneider and Pozzi, 2011*; *Chun and Surh, 2004*; *Smith et al., 2000*). Unlike uniformly expressed COX-1, constitutive COX-2 expression is normally restricted to a few organs but can be induced by a variety of stimuli including cytokines, oncogenes, growth factors, and hormones (*Chun and Surh, 2004*; *Smith et al., 2000*). Induced overexpression of COX-2 and PGs has been observed with varied frequency in a number of human cancers, including breast, colon, prostate, and lung (*de Moraes et al., 2007*). Numerous genetic and correlation studies have documented the causal involvement of COX-2 in tumor development (*Gupta et al., 2007*; *Liu et al., 2001*; *Minn et al., 2005*; *Oshima et al., 1996*; *Markosyan et al., 2011*).

Induction of COX-2 expression can be mediated by a number of intracellular signaling pathways in different cell types (*Smith et al., 2000*). Indeed, involvement of PKC, Ras, and Wnt pathways through the

activation of MAPK kinase family of proteins including ERK, JNK and p38 in a cell type-specific manner has been demonstrated (*Gauthier et al., 2005*; *Wang et al., 2001*; *Rodriguez-Barbero et al., 2006*; *Ramsay et al., 2003*; *Zhai et al., 2010*; *Li et al., 2009*). DNA binding sites for mediators of these signaling events, such as transcription factors NF-κB, AP1, CREB C/EBP, NF-IL6, MEF2, and TCF4/LEF1 have been identified in the COX-2 promoter and further studies demonstrated their functional involvement in regulating COX-2 transcription in a variety of experimental systems (*Chun and Surh, 2004*; *de Moraes et al., 2007*).

In this study, we describe a novel layer of *COX-2* transcriptional regulation in cells relevant to human tumorigenesis. Using primary human mammary epithelial cells (HMECs) and a PMA-driven human monocyte-macrophage differentiation system, we demonstrate that a nuclear antisense long non-coding RNA PACER (**P**50-**A**ssociated **C**OX-2 **E**xtragenic **R**NA) is expressed in the upstream region of COX-2 and functions to directly sequester the repressive NF-κB p50 subunit from the COX-2 promoter. This facilitates the recruitment of p300 histone acetyltransferase and a domain-wide increase in chromatin accessibility, as well as assembly of RNA Polymerase II pre-initiation complexes. We also show that PACER expression is controlled by two CTCF/cohesin complexes assembled in the 5' UTR region and at a distal upstream site. CTCF/cohesin induces lncRNA expression by establishing a chromatin domain marked by increased H3K4 methylation, H4K8 acetylation and decreased H4K20 trimethylation, thus creating a permissive chromatin environment and protecting the COX-2 regulatory region from surrounding repressive chromatin. These findings offer unexpected insights into the mechanisms of gene regulation by CTCF/cohesin and by lncRNA-mediated repressor eviction.

## Results

### Identification of an antisense long non-coding RNA in the upstream region of the human *COX-2* gene

We initially hypothesized that *COX-2* expression might be regulated by a non-coding antisense RNA overlapping the gene or its promoter. Our search failed to identify lncRNAs within the COX-2 coding region, so we examined the region upstream of the COX-2 transcription start site. This region contains numerous putative binding sites for transcription factors but no additional annotated genes. We performed chromatin immunoprecipitation (ChIP) experiments using antibodies directed against RNA Polymerase II (RNAP II), which revealed high levels of RNAP II as far as ~1 kb upstream of the COX-2 transcribed region. Intriguingly, we detected a peak of RNAP II binding at approximately −0.45 kb (*Figure 1A*). RNAP II molecules in this region contained C-terminal domain (CTD) phosphorylation on serine 5 as demonstrated by ChIPs with antibodies recognizing Ser-P RNAP II specifically (*Figure 1B*). These findings suggested that an additional RNAP II promoter was located ~300 bp upstream of the COX-2 promoter, potentially driving expression of an unidentified species of extragenic RNA.

We then examined whether any transcripts were synthesized within this region using quantitative RT-PCR (RT-qPCR). Primer walking indicated that the COX-2 upstream region is transcriptionally active roughly from −0.3−0.4 kb to approximately −1.5−2 kb (*Figure 1C*). To determine the directionality of transcription, we used strand specific RT-qPCR. When cDNA synthesis was primed with either of the 'sense' oligos (which detect antisense transcripts), contiguous transcripts were detected extending roughly up to −0.3−0.4 kb (*Figure 1D*). By contrast, 'anti-sense' oligos did not produce detectable signals within the upstream region (*Figure 1E*) in contrast to positive amplification within the coding region (*Figure 1F*). These experiments document the existence of a contiguous antisense transcript originating at approximately −0.3 kb upstream of the COX-2 mRNA start site. To determine the exact 5' and 3' ends of the transcript, rapid amplification of cDNA ends (RACE) was employed (*Figure 1G*, *Figure 1—figure supplements 1, 2*). 3' RACE repeatedly showed that a polyA tail is added after position −1022, consistent with the presence of a classical polyadenylation signal AAUAAA at position −999 (*Figure 1—figure supplement 1*). Sequencing of the 5' ends showed a variety of positions (from −257 to −332) indicating that alternative start sites are used by RNA Polymerase (*Figure 1—figure supplement 2*). This is consistent with the absence of a TATA box around the transcription start site of this novel RNA. Given that our qPCR analysis showed the presence of transcripts extending up to −1.5−2 kb upstream, we cannot exclude the existence of additional species of ncRNA within the COX-2 upstream region. Exploring the function of those molecules could be a subject of a future study.

A survey of a recent study utilizing strand specific Global Run-On sequencing (GRO-seq) confirmed that the upstream region of COX-2 is indeed transcribed in an antisense orientation (*Galbraith et al., 2013*).

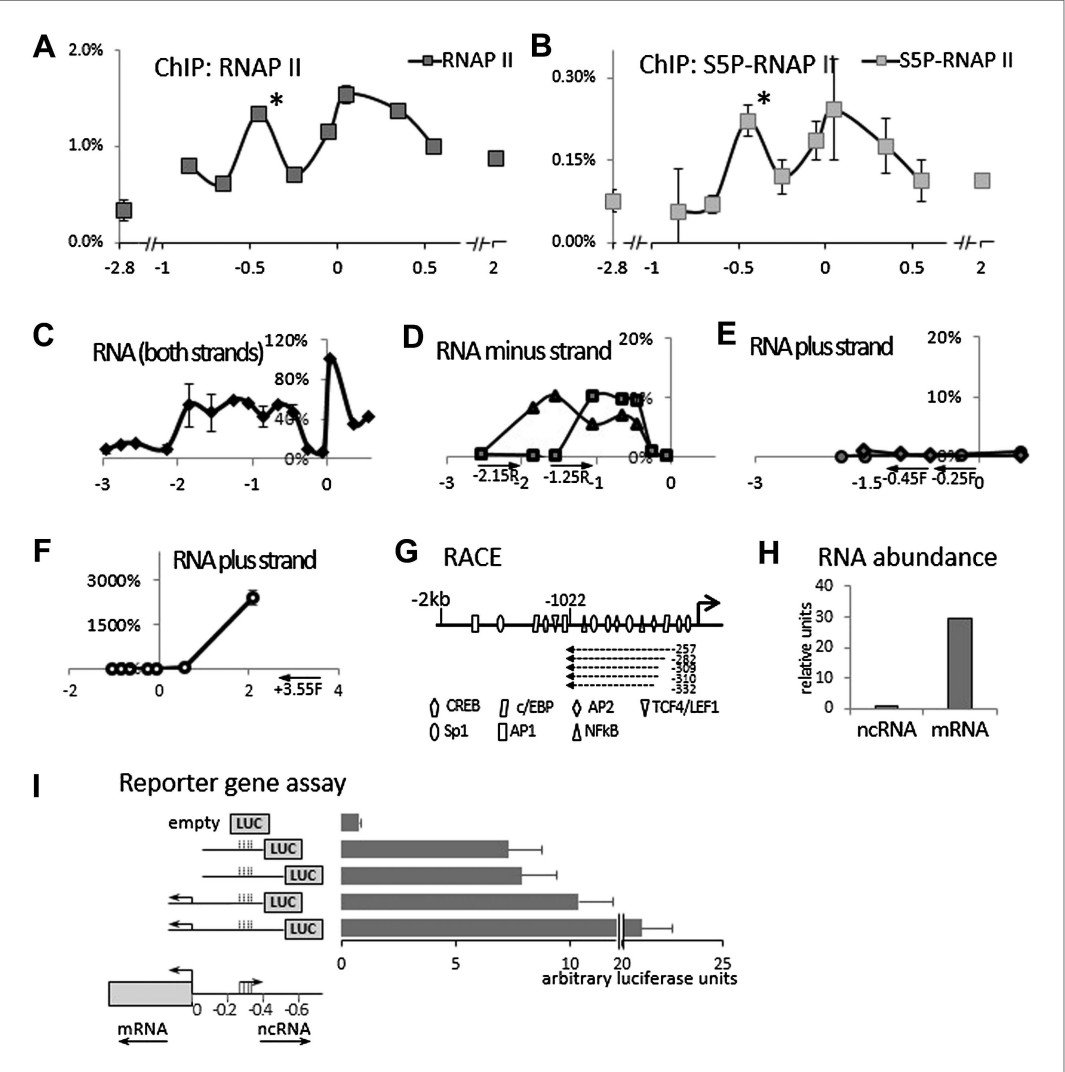

**Figure 1**. Identification of an antisense long non-coding RNA in the upstream region of the *COX-2* gene. (**A** and **B**) ChiP experiments were performed with chromatin extracts from HMEC cells with antibodies directed against RNAP II or phospho-serine 5 modified RNAP II, respectively, and analyzed by qPCR. Each histogram represents one qPCR data point (amplicon). Non-COX-2 promoter RNAP II peak is marked with an asterisk. X-axis scale is relative to COX-2 transcription start site. (**C**) Transcript abundance in the *COX-2* locus in HMECs was measured by RT-qPCR. cDNA was primed with random oligos and therefore provide no information on strand specificity. Each diamond represents one RT-qPCR data point (amplicon). Y-axis values are relative to the signal obtained with the 5' UTR amplicon. (**D** through **F**) Strand specific RT-qPCR. cDNA synthesis was performed with two sense oligos (m2R and jR) measuring antisense transcription or three antisense oligos (jF, eF, and control −3.55F) measuring sense transcription. (**G**) A schematic representation of lncRNA ends mapped with RACE. (**H**) Relative abundance of COX-2 mRNA and non-coding RNA. (**I**) A reporter gene assay used to measure the strength of the ncRNA promoter. Luciferase-only construct (empty) or constructs containing indicated fragments of COX-2 upstream region were transfected into 293T cells and luciferase activities were measured 48 hr post-transfection. Activity units reflect the ratios between Firefly and control Renilla luciferases.

The following figure supplements are available for figure 1:

**Figure supplement 1**. Mapping the 3' end of the non-coding RNA transcript within the COX-2 upstream region by RACE.

**Figure supplement 2**. Mapping the 5' end of the non-coding RNA transcript within the COX-2 upstream region by RACE.

*Figure 1. Continued on next page*

*Figure 1. Continued*

**Figure supplement 3**. Entire sequence of PACER, corresponding to the longest 5' variant mapping to nucleotides −257 to −1022.

**Figure supplement 4**. GRO-seq data supports the existence of a long non-coding RNA species within COX-2 upstream region.

A strong GRO-seq signal extends up to approximately −1100 bp upstream of COX-2 transcription start site, and therefore correlates with our finding of a major lncRNA species in this region (*Figure 1—figure supplement 4*). We used qPCR to accurately measure the relative abundance of the newly identified ncRNA and COX-2 mRNA. To do this, pairs of oligos mapping within one exon of the mRNA and within the intronless ncRNA were used for cDNA amplification and their abundance was measured relative to reference genomic DNA. The results indicate that the mRNA is present in ~30-fold molar excess as compared to the ncRNA (*Figure 1H*).

To corroborate the finding that two transcriptional units possessing separate RNAP II promoters exist within the COX-2 locus, we used reporter gene assays (*Figure 1I*). 400 or 500 bp fragments of DNA upstream of the COX-2 mRNA start site containing the putative ncRNA promoter was fused to the Luciferase gene (constructs 2 and 3). The activities of these constructs were compared to analogous fusions containing additional sequence spanning the COX-2 mRNA promoter (constructs 4 and 5). Constructs containing the ncRNA promoter but lacking the mRNA promoter were highly active as compared to the promoterless empty plasmid. Addition of the mRNA promoter increased the activity of the constructs (up to twofold), indicating that the two regulatory elements could exhibit synergistic behavior. Nevertheless, these experiments show that a bona fide promoter driving the expression of ncRNA is present in the region ~300–400 bp upstream of the mRNA transcription start site.

Bioinformatic analysis of the nucleotide sequence failed to support the hypothesis that the novel RNA encodes a functional protein. The longest detected Met-initiated putative ORF is 43 aa and this peptide contains no identifiable protein domains and bears no similarity to any protein in protein databases (not shown). In addition, polysome profiling indicated that the ncRNA does not co-sediment with the poly-ribosome fraction (D Henderson, personal communication). While we cannot exclude that the RNA might encode a small peptide of unknown function, we conclude with high confidence that it represents a novel species of long non-coding RNA (lncRNA). We will continue to refer to it as **P**50-**A**ssociated **C**OX-2 **E**xtragenic **R**NA (PACER, see below). The entire nucleotide sequence of PACER is shown in *Figure 1—figure supplement 3*.

During the preparation of this manuscript, Carpenter and colleagues reported that a species of lncRNA located ~50 kb downstream of the mouse COX-2 gene affects expression of many immune-related genes both positively and negatively, likely through interactions with RNA binding proteins hnRNP A/B and hnRNP A2/B1 in murine macrophages (*Carpenter et al., 2013*). However, COX-2 mRNA expression was neither affected by knockdown or overexpression of this lncRNA; nor in the initial screen which identified a group of lncRNAs induced by treatment with the Tlr2 ligand Pam3CSK4. Therefore, the name of this lncRNA (lincRNA-COX-2) merely reflects its genomic localization, not its functional relationship with the COX-2 gene, unlike the lncRNA reported here.

## CTCF binding demarcates an active regulatory domain in the COX-2 upstream region

Binding sites for known mediators of COX-2 activation, such as NF-κB, AP1, NF-IL6, CREB, TCF4/LEF1, cluster between the COX-2 transcription start site and approximately −1.5 kb upstream of the promoter (*Figure 1G*) (*de Moraes et al., 2007*). This clustering, together with the presence of lncRNA suggests the existence of a specialized chromatin domain encompassing all elements involved in COX-2 expression. Upon inspection of genomic databases, we noticed that the region far upstream (-3kb to > -50 kb) of COX-2 is densely populated by repetitive DNA elements, whereas the region from +1 to roughly −3 kb upstream of the promoter is devoid of them (*Figure 2A*). We hypothesized that CTCF, the only known mammalian insulator/boundary factor, could be involved in regulation of COX-2 expression by establishing a boundary between the region containing repetitive DNA and the proximal promoter.

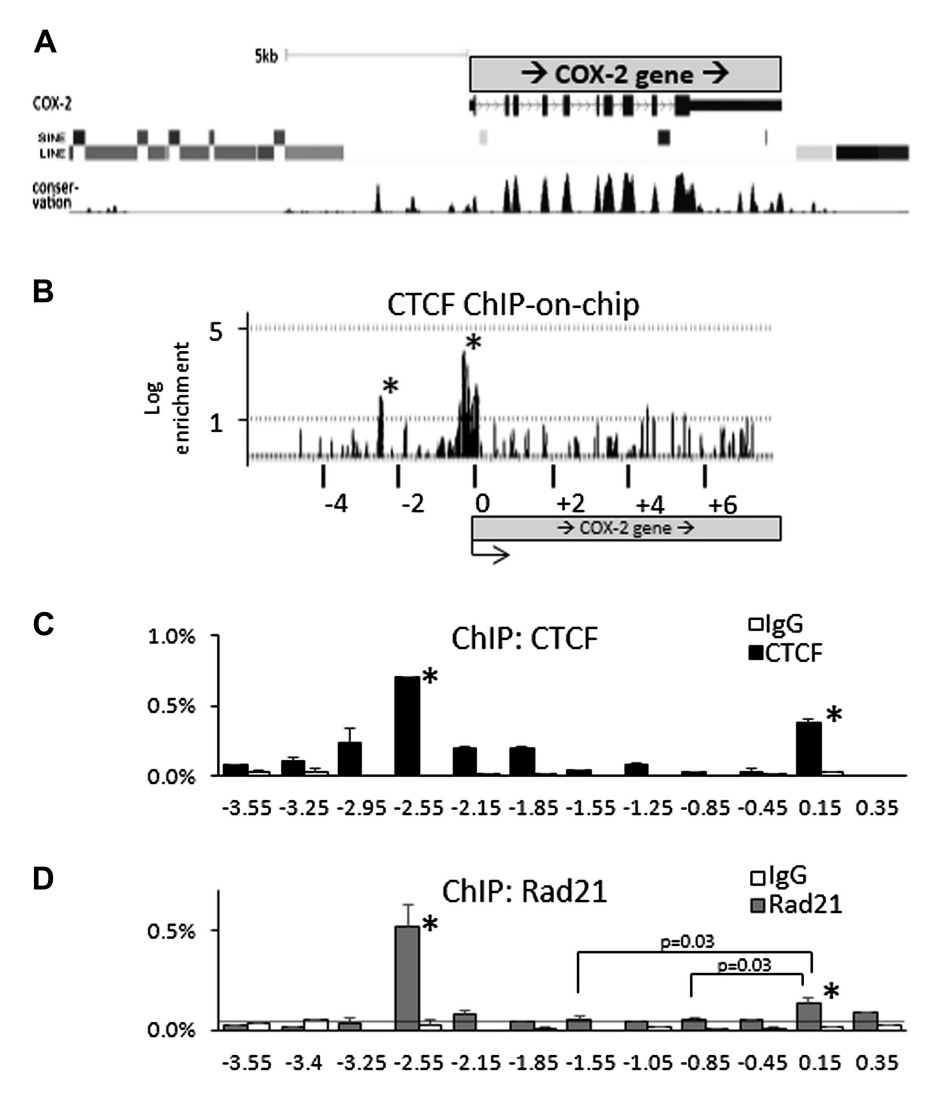

**Figure 2**. CTCF/cohesin complexes bind to two sites within the *COX-2* locus, encompassing the promoters of mRNA and lncRNA. (**A**) Genomic neighborhood of the human *COX-2* gene. Note the repeat-DNA-free domain extending up to approximately −3 kb upstream of the COX-2 promoter. (**B**) ChIP-on-chip analysis of CTCF binding within the *COX-2* genomic domain in human mammary epithelial cells (HMECs). Two CTCF peaks are evident at approximately −2.5 kb and the proximal promoter. Numerous other loci on the array served as negative controls (***Figure 2—figure supplement 1***). Results are represented as log2ratios between hybridization signals obtained with CTCF-ChIP and input DNA samples. Each vertical line represents one probe on the array. X-axis kb scale is relative to the COX-2 transcription start site. Significant signals are marked by asterisks. (**C** and **D**) Mapping of CTCF and cohesin binding in HMECs analyzed by ChIP-qPCR primer-walking. Amplicon coordinates are relative to the COX-2 transcription start-site.

The following figure supplements are available for figure 2:

**Figure supplement 1**. Selected results of CTCF ChIP-on-chip experiments.

**Figure supplement 2**. Identification and characterization of the 5′ UTR CTCF binding site in the *COX-2* gene.

**Figure supplement 3**. Mutations in predicted CTCF sites abolish CTCF recruitment to DNA templates.

We addressed this possibility using ChIP-on-chip experiments. A custom genomic DNA tiling array was designed to include the entire human *COX-2* gene and 50 kb flanking sequence on each side. Numerous other genes were also included on the array (data not shown). Chromatin immunoprecipitation

(ChIP) experiments were performed with extracts from primary human mammary epithelial cells (HMECs) using antibodies directed against CTCF. IP DNA was purified, amplified, and hybridized to DNA arrays. Before hybridization, enrichment of known genomic CTCF binding sites in IP DNA was verified by qPCR.

As expected, we found CTCF at numerous loci across the regions included on the array (*Figure 2—figure supplement 1*). The locations and patterns of binding were largely similar to those seen in previously studied cells (data not shown) (*Kim et al., 2007*; *Xie et al., 2007*; *Barski et al., 2007*). We found two CTCF sites in the *COX-2* locus, one located in the promoter region and the other at approximately −2.5 kb, precisely at the interface between the region of repetitive DNA and the putative COX-2 regulatory region (*Figure 2A,B*). Computer prediction revealed the presence of a very well-conserved CTCF recognition motif GCAGCAGAAGGGGGCAGTA at position −2508 and a weaker one GCAGCGCCTCCTTCAGCTCCA at position +65. EMSA and DNA recruitment assays provided additional evidence that CTCF indeed binds at +65bp relative to transcription start site (*Figure 2—figure supplements 2, 3*).

Previous studies have shown that the majority of CTCF-bound genomic sites are co-occupied by cohesin, a protein that has diverse roles in chromosome biology including regulating sister chromatid cohesion and DNA topologies (*Parelho et al., 2008*; *Rubio et al., 2008*; *Stedman et al., 2008*; *Wendt et al., 2008*). Together, CTCF and cohesin form a complex that regulates a variety of chromatin-related functions such as inter-chromosomal interactions, enhancer blocking and insulation (*Merkenschlager and Odom, 2013*). We thus investigated whether cohesin and CTCF bind to the same locations within the COX-2 upstream region in HMECs, using ChIP-qPCR with antibodies recognizing CTCF and cohesin subunit Rad21 (*Figure 2C,D*). We found that CTCF and cohesin strongly bind to the −2.5-kb site, whereas the promoter site is also recognized by both proteins, albeit to a lesser extent, especially in the case of Rad21. ChIP-qPCR experiments with other known CTCF interactors, such as B23/nucleophosmin, Topoisomerase IIβ, PARP-1 and, C23/nucleolin, failed to reveal specific binding in the COX-2 upstream region (data not shown).

The relative CTCF ChIP signal strengths at the COX-promoter and at the −2.5 kb site seem to be different in ChIP-on-chip experiments as compared to ChIP-qPCR. We tend to believe that the qPCR reflects the actual situation, because of its quantitative nature, as opposed to hybridization-based detection in the case of ChIP-on-chip. The most probable cause of the discrepancy is the fact that the COX-2 promoter region contains a CpG island extending from position +245 to −525, and thus probes covering this region contained an unusually high GC content, which could skew hybridization efficiency producing aberrantly strong and wide signal peak.

## CTCF/cohesin complexes control a chromatin domain and induce expression of a long non-coding RNA and COX-2 mRNA

Given that CTCF/cohesin occupies thousands of sites across the genome, we verified their functional involvement in regulating *COX-2* expression. To this end, we used small molecule RNA interference to transiently ablate the expression of CTCF or cohesin in HMECs. CTCF knockdown reduced COX-2 mRNA to ~20%, and the COX-2 protein to almost undetectable levels (*Figure 3A,B*). These effects appear to be direct, because not only CTCF mRNA and protein levels were decreased by siRNAs (*Figure 3B,C*) but also CTCF binding to both promoter and distal −2.5 kb sites in the *COX-2* locus (*Figure 3D*). Similar to CTCF, siRNA-mediated knockdown of the cohesin subunit SMC3 reduced COX-2 mRNA levels, albeit to a lesser extent (*Figure 3A*). Importantly, knockdown of CTCF-reduced expression of extragenic RNA to practically undetectable levels (*Figure 3E*). This means that CTCF exerts more global control over the *COX-2* locus by regulating both mRNA and ncRNA expression from two distinct promoters. This prompted us to investigate whether CTCF controls COX-2 expression by creating a specialized chromatin neighborhood within the putative CTCF/cohesin chromatin loop. To do this, we performed ChIP-qPCR experiments to analyze the chromatin modification status in the COX-2 promoter region. Interestingly, the effects of CTCF knockdown on several activation and repression-associated histone marks extended to ~2.5 kb upstream. High levels of H3K4me2 and me3 showed a local maximum at approximately −0.7 kb and were reduced twofold to threefold across the entire ~3 kb domain in the absence of CTCF (*Figure 3F,G*). H4K8 acetylation levels extended even further upstream, and were reduced significantly by CTCF knockdown (*Figure 3H*). A repressive mark H4K20me3 had a more localized pattern, with two discernible peaks at −2.5 kb and approximately −1.2 kb. This modification was increased at −2.5 kb upon CTCF knockdown (*Figure 3I*). No significant changes were observed for H3K9me3 (*Figure 3J*), or H4K12-Ac, H2AZ, and unmodified H3 (data not

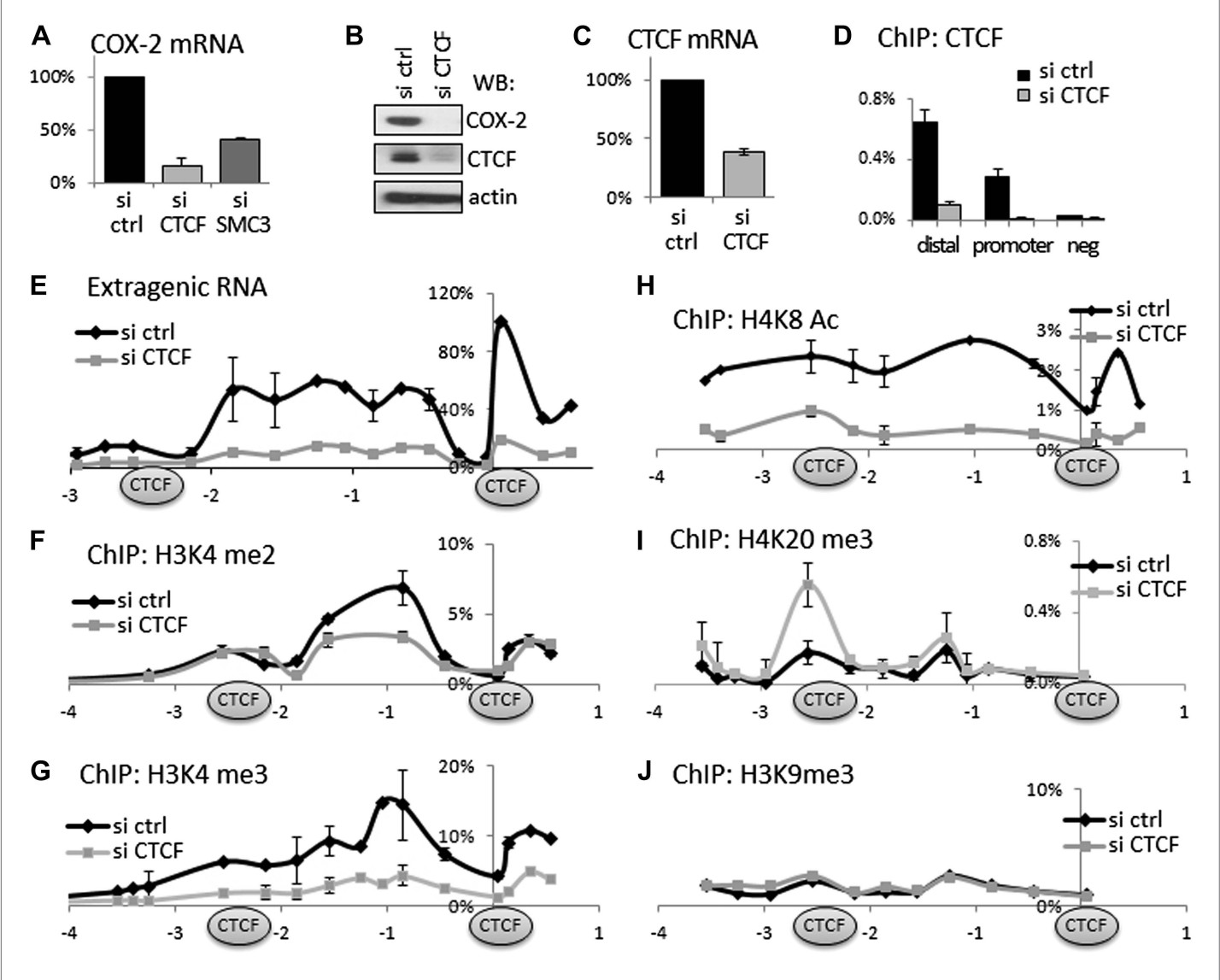

**Figure 3**. CTCF and cohesin maintain COX-2 mRNA and PACER lncRNA expression by demarcating a chromatin domain that is characterized by decreased H4K20 trimethylation, increased H3K4 di- and tri- methylation and increased histone acetylation. (**A**) siRNA-mediated knockdown of CTCF or a cohesin subunit SMC3 reduces COX-2 mRNA levels in HMECs. RT-qPCR was performed with RNA prepared from HMECs 72 hr after transfection with the indicated siRNAs. Signals were normalized with 18S rRNA, TBP and GAPDH genes. (**B**) Western blot analysis of COX-2 and CTCF levels upon siRNA-mediated knockdown of CTCF. (**C**) CTCF mRNA levels in control and siRNA-CTCF transfected HMECs. (**D**) Binding of CTCF to the COX-2 promoter and distal sites were analyzed by ChIP-qPCR in control and CTCF knockdown cells. (**E**) Transcription across the *COX-2* locus was measured by RT-qPCR in control (black diamonds) or siCTCF-transfected HMECs (grey squares). Signals were normalized as above and re-normalized using genomic DNA. (**F–J**) Levels of activation-associated (H3K4me2 and me3, H4K8Ac) and repression-associated (H4K20me3, H3K9me3) histone modifications were measured using ChIP-qPCR in HMECs transfected with control siRNAs (black lines) or siRNA against CTCF (grey lines). Each diamond or square denotes a qPCR amplicon. CTCF/cohesin binding sites are marked with oval shapes. X-axis scale is relative to the COX-2 transcription start site.

shown). Taken together, these experiments indicate that CTCF regulates the *COX-2* locus by establishing and maintaining an open chromatin domain demarcated by two CTCF/cohesin complexes; inducing extragenic transcription of PACER within this domain; and activating COX-2 mRNA expression.

## A long non-coding antisense RNA controls COX-2 mRNA transcription

Given that the mechanism of gene activation by CTCF/cohesin is not fully understood and our observation that CTCF binding to the *COX-2* locus is required for both PACER and COX-2 mRNA

expression, we wondered whether PACER might serve as an intermediary in activating mRNA transcription. To explore this, we first performed transient knockdown experiments using siRNA specific to the PACER sequence. Indeed, siRNA-mediated reduction of PACER decreased COX-2 mRNA levels to a comparable extent as observed with CTCF knockdowns in HMECs (*Figure 4A*) and COX-2 protein was reduced to negligible levels (*Figure 4B*). Similar results were obtained using another siRNA specific to PACER (*Figure 4—figure supplement 1*), confirming siRNA specificity. Importantly, the effects of PACER knockdown were independent of CTCF since CTCF protein levels remained unchanged (*Figure 4B*) and CTCF binding to the *COX-2* locus was not reduced (*Figure 4—figure supplement 2*).

So far, we focused our analysis of the *COX-2* locus in human mammary epithelial cells because these cells have been implicated in breast tumorigenesis (*Crawford et al., 2004*). However, COX-2 is also expressed by tumor-associated macrophages (TAMs), which are known to accelerate tumor formation and metastasis in breast and other cancer types (*Pollard, 2004*; *Chen and Smyth, 2011*). We therefore decided to further examine lncRNA-mediated regulation of *COX-2* expression in the human monocyte/macrophage cell line U937, as these cells offer unique advantages as a model system to study gene regulation. U937 monocytes can be differentiated into macrophages with phorbol myristate acetate (PMA) and *COX-2* expression can be efficiently induced in U937 macrophages with a variety of stimuli, including LPS (schematically shown in *Figure 4C*) (*Arias-Negrete et al., 1995*).

We first established stable U937 lines carrying short hairpin RNAs (shRNA) targeted against PACER using lentiviral vector delivery. U937-sh-lncRNA monocytes showed no detectable COX-2 mRNA or lncRNA expression, as did control monocytes carrying shRNA directed against the Luciferase gene (*Figure 4D,E*). Upon PMA-induced differentiation, both COX-2 mRNA and PACER expression were strongly induced in control U937 cells. By contrast, in PACER-knockdown cells, PMA induced lncRNA expression to much lower levels compared to control cells, and, as a result, COX-2 mRNA levels were also markedly reduced (*Figure 4D,E*). Induction of PMA-differentiated macrophages with LPS greatly up-regulated *COX-2* expression in control macrophages over a 6-hr time course, with the highest levels of expression reaching a plateau at ~2 hr post induction (*Figure 4H*). In knockdown cells, the effect of LPS was severely attenuated, with COX-2 mRNA reaching ~60% of maximal control levels 4 hr post-induction, but not abolished, consistent with the fact that PACER expression was also up-regulated during LPS treatment (*Figure 4G*). Finally, Western blotting in whole cell extracts confirmed PACER-mediated effects on COX-2 protein levels both in PMA differentiation (*Figure 4F*) and during LPS induction (*Figure 4I*). Interestingly, PACER knockdown appears to greatly reduce COX-2 protein levels upon PMA stimulation of monocytes but only transiently affect COX-2 levels upon LPS-induction. This suggests that PACER plays an essential role early in *COX-2* gene activation during monocyte differentiation into macrophages by controlling the critical transition from a transcriptionally inactive locus to one competent for basal transcription.

We analyzed the relative abundance of PACER within the cell using biochemical fractionation of U937 macrophages followed by qPCR. The vast majority (more than 75%) of the lncRNA was found in the nucleus, with one third being stably integrated into chromatin and the rest occupying nuclear space; the nuclear fraction increased even more upon LPS stimulation (*Figure 4J*). This distribution is very different from that of COX-2 mRNA, of which around 50% is found in the cytoplasm. This observation suggests that PACER performs its functions in the nucleus. Small amounts of PACER found in the cytoplasm likely reflect a minor leakage of nucleoplasm into the cytoplasmic fraction, as the same is observed for control intronic sequences (*Figure 4J*).

## PACER lncRNA facilitates p300 binding, chromatin opening, RNA polymerase II recruitment, and transcriptional activation

We exploited the U937 differentiation system to gain more insight into the mechanisms by which PACER regulates basal and induced *COX-2* transcription through chromatin modification status, transcription factor interaction, and co-activator recruitment. First, we analyzed binding of CTCF, cohesin, Sp1 and AP1 to the COX-2 upstream region upon PACER knockdown. Sp1 and AP1 have been previously implicated in mediating *COX-2* activation in various cell types, including macrophages (*Xie and Herschman, 1996*; *Xu et al., 2000*). We found that PACER knockdown did not significantly affect binding of any of these four factors to the COX-2 promoter region, apart from a modest decrease of the AP1 component c-Jun binding to distal (−0.45 kb and −0.85 kb) sites but not to the COX-2 proximal promoter (*Figure 5—figure supplement 1*).

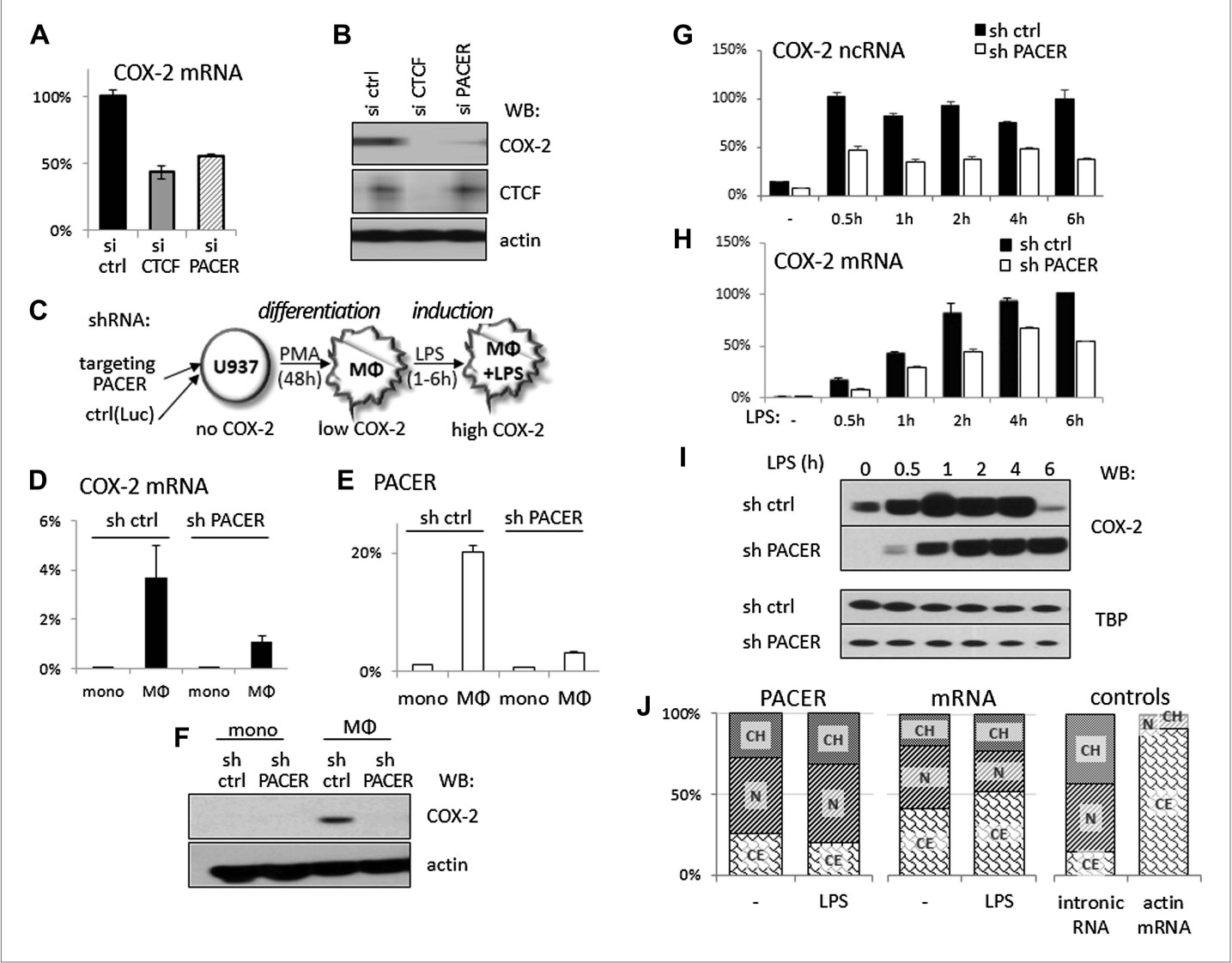

**Figure 4**. The antisense PACER lncRNA is a positive regulator of COX-2 expression in human mammary epithelial cells and in monocyte-derived macrophages before and after LPS stimulation. Levels of COX-2 mRNA (**A**) and protein (**B**) were measured by RT-qPCR and Western blotting in control-scrambled siRNA, siRNA-CTCF, and siRNA-lncRNA transfected HMECs 72 hr post transfection. (**C**) Schematic representation of the monocyte-macrophage system used in this study. The human monocyte cell line, U937, was differentiated to macrophages with PMA and induced to express high levels of COX-2 by LPS stimulation. (**D** and **E**) COX-2 mRNA (**D**) and PACER (**E**) levels were measured by RT-qPCR in U937 lines carrying stably integrated control shRNA or shRNA-PACER expressing constructs before and after differentiation into macrophages. Values were normalized using 18S rRNA, TBP, and ß-globin genes. (**F**) COX-2 protein expression analyzed by Western blotting in the same experiment. (**G** and **H**) COX-2 ncRNA and mRNA levels were measured in a time course of LPS stimulation in control U937 (black bars) and PACER knockdown U937 macrophages (white bars). Values are relative to maximal COX-2 expression in control U937 cells after 6 hr LPS stimulation. (**I**) COX-2 protein expression analyzed by Western blotting in the same experiment. TBP served as a control. (**J**) Subcellular localization of COX-2 mRNA, PACER lncRNA and controls: nuclear intronic RNA and exclusively cytoplasmic actin mRNA. Relative levels in each fraction were measured by RT-qPCR and plotted such that they add up to 100%. CH, chromatin-bound fraction, N, nucleoplasm, CE, cytoplasm.

The following figure supplements are available for figure 4:

**Figure supplement 1**. Knockdown of COX-2 mRNA with siRNA targeting PACER (si+753 and si+870) or positioned outside PACER (+1049).

**Figure supplement 2**. Binding of CTCF to the COX-2 promoter upon PACER knock-down in HMECs.

We next analyzed the effects of PACER knockdown on chromatin modifications within the COX-2 upstream domain. ChIP-qPCR experiments showed no significant changes in H3K4 trimethylation or H2A.Z association (data not shown) and a modest effect on H3K4 dimethylation (*Figure 5—figure supplement 2*). By contrast, histone H3 acetylation was clearly affected by the loss of PACER. Levels of H3Ac in non-induced macrophages were considerably reduced at −1.25 kb and −2.5 kb in shPACER-cells and the LPS-induced increase in H3Ac was markedly attenuated compared to control macrophages (*Figure 5A*). Even more striking effects were evident for H4 acetylation. H4Ac was induced in response to LPS treatment across the upstream COX-2 domain extending to approximately −1.3 kb. Knockdown of PACER abolished H4Ac to practically background levels (*Figure 5B*). These experiments demonstrate that PACER exerts its function through modulation of chromatin acetylation rather than methylation, as has been shown previously for several lncRNAs (*Wutz et al., 2002*; *Nagano et al., 2008*; *Khalil et al., 2009*; *Zhao et al., 2010*; *Tsai et al., 2010*; *Yap et al., 2010*; *Wang et al., 2011*).

We sought to identify histone acetyltransferase complexes that could mediate PACER function. While HAT activities have not so far been implicated in the function of known lncRNAs, the p300 histone acetyltransferase has been shown to participate in COX-2 induction in human macrophages (*Xiao et al., 2011*; *Kang et al., 2006*). To examine the possibility that COX-2 lncRNA might affect recruitment of p300 to the *COX-2* locus, we performed ChIP-qPCR experiments. p300 was absent from the COX-2 promoter in non-induced macrophages but was recruited to the promoter upon LPS stimulation (*Figure 5C*). Primer walking identified a major binding site at approximately −0.25 kb. Importantly, in shPACER knockdown macrophages, LPS-induced p300 association was severely decreased suggesting that the lncRNA regulates *COX-2* expression through p300 recruitment and histone acetylation.

To explore the possibility that PACER could affect (directly or indirectly), the assembly of RNA Polymerase II preinitiation complexes and/or later steps in transcriptional activation, we used ChIP-qPCR to assess the association of bulk RNAP II and RNAP II modified at serine 5 or serine 2 within its C-terminal domain across the entire *COX-2* locus. Upon PACER knockdown, association of RNAP II was markedly reduced at both the mRNA promoter (amplicons −0.05, 0.05) and PACER promoter (−0.25, −0.45), as well as across the *COX-2* gene (0 to 4 kb) and PACER gene (- 1 kb to −0.25 kb) (*Figure 5D*). Elongating/terminating S2-modified polymerase association, which is usually skewed toward the ends of transcribed regions (6 kb for mRNA, −1 kb for lncRNA in this case), was also markedly lower in KD cells (*Figure 5E*). Finally, early initiating/paused S5-modified polymerase, usually enriched within the first 50 bp of the transcribed region (0 for mRNA, −0.25 to −0.45 kb for lncRNA here) was also significantly decreased upon PACER knockdown (*Figure 5F*). These results demonstrate that PACER influences early steps in transcription initiation at the stage of formation of RNAP II preinitiation complexes, likely through p300-induced changes in promoter-associated histone acetylation.

## PACER directly interacts with the repressive NF-κB subunit p50 to occlude it from the COX-2 promoter

How does PACER recruit RNAP II initiation complexes and p300 to the COX-2 promoter? To explore the possibility that the lncRNA might directly interact with p300, we carried out RNA immunoprecipitation (RIP) experiments. Antibodies against p300 were incubated with extracts containing intact cellular RNAs, RNA was then purified from immunoprecipitates and analyzed by RT-qPCR. These experiments failed to detect the presence of lncRNA in p300 RIP samples (*Figure 6A*). Similarly, p300 was not detected in RNA pulldown experiments, where we used lncRNA fragments to isolate interacting proteins from whole cell extracts (data not shown). We therefore conclude that PACER does not directly bind to p300.

LPS induces several signaling pathways in macrophages, which predominantly converge on activation of NF-κB and its target genes, including *COX-2*. NF-κB-mediated induction of *COX-2* has been previously shown to involve p300 recruitment to DNA (*Xiao et al., 2011*; *Kang et al., 2006*). This raised the possibility that PACER could interact with a member of the NF-κB family. Indeed, RIP experiments demonstrated that p50, the small subunit of NF-κB, interacts directly with the lncRNA (*Figure 6A,B*). We also detected p50 in RNA pulldown experiments using the COX-2 lncRNA but not its inverted control (data not shown). By contrast, PACER did not interact with the NF-κB large subunit p65/RelA by RIP or with CTCF and control SNRNP70 proteins (*Figure 6A*). Binding of p50 to the lncRNA could be detected using amplicons located at multiple positions within the PACER sequence (*Figure 6B*), demonstrating that

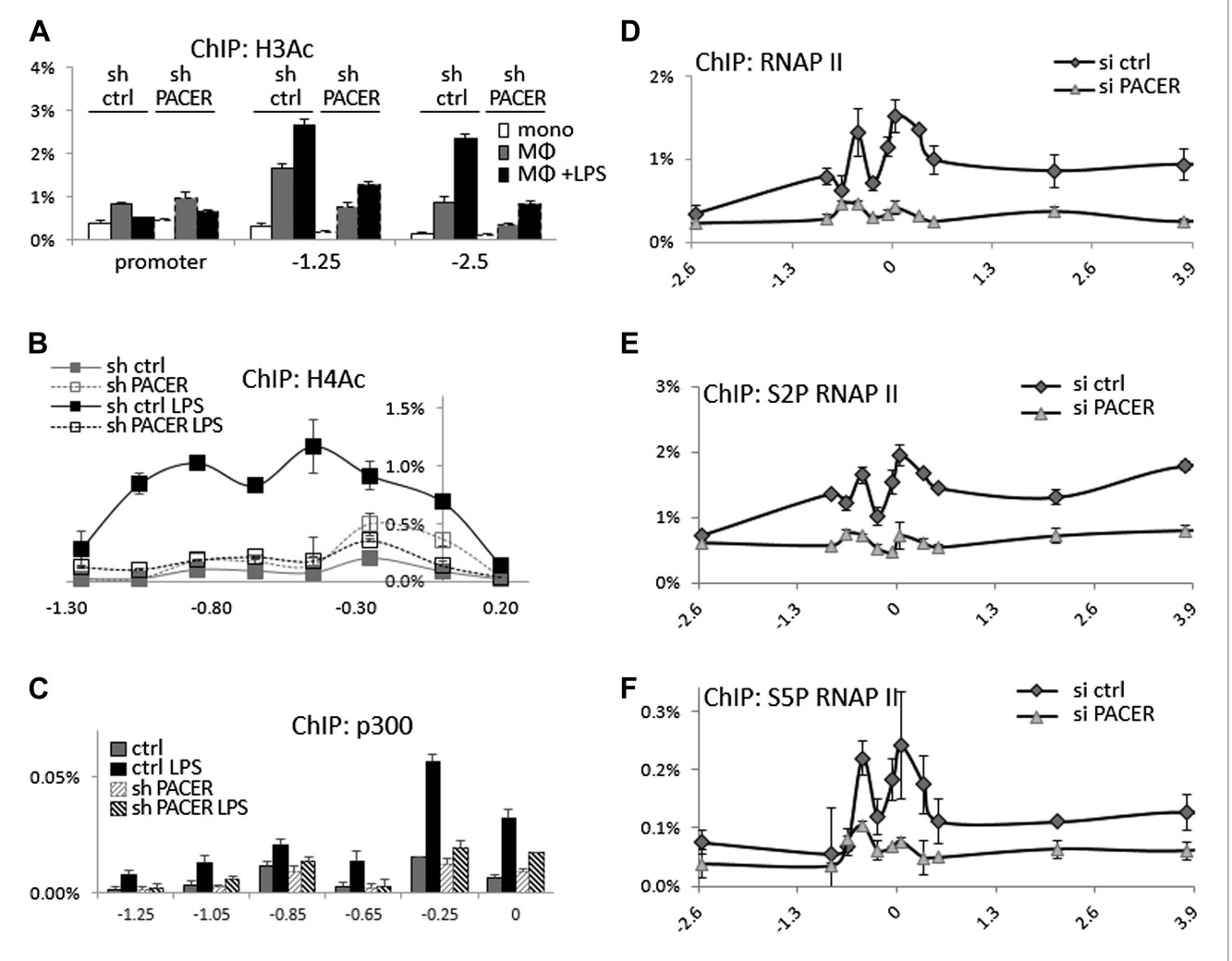

**Figure 5**. PACER facilitates recruitment of p300 HAT and RNAP II pre-initiation complexes to increase histone acetylation and induce *COX-2* transcription. (**A**) ChIP-qPCR was used to measure levels of histone H3 acetylation in control U937 and sh-lncRNA U937 monocytes (white bars), macrophages (grey-bars) and LPS-stimulated macrophages (black bars) at the COX-2 promoter, −1.25 kb upstream and −2.5 kb upstream of the COX-2 transcription start site. (**B**) ChIP-qPCR analysis of histone H4 acetylation across the COX-2 upstream region in control and sh-PACER U937 macrophages before and after LPS stimulation. (**C**) Association of p300 with the COX-2 upstream region was analyzed by ChIP-qPCR in control (filled squares) and sh-PACER (open squares) U937 macrophages before and after LPS stimulation. (**D**–**F**) Association of bulk RNAP II (**D**), S2-phosphorylated (**E**) and S5-phosphorylated (**F**) RNAP II across the *COX-2* locus analyzed by ChIP-qPCR in control HMECs (diamonds) or PACER-knockdown (triangles) HMECs. Association of RNAP II with the *COX-2* locus in control or siRNA-PACER transfected HMECs was assayed using ChIP-qPCR 72 hr post transfection. X-axis scale is relative to the COX-2 transcriptional start site.

The following figure supplements are available for figure 5:

**Figure supplement 1**. Binding of c-Jun to the COX-2 region upon PACER knock-down in U937 monocytes, macrophages, and LPS-stimulated macrophages analyzed by ChIP.

**Figure supplement 2**. Levels of H3K4 dimethylation at COX-2 region upon PACER knock-down in U937 monocytes, macrophages, and LPS-stimulated macrophages analyzed by ChIP.

p50 interacts with full length, intact PACER. Finally, binding to PACER was specific, because p50 did not interact with control U1 snRNA (***Figure 6C***). Given these results, we investigated whether association of p50 with the COX-2 promoter was affected by knockdown of PACER using ChIP-qPCR. We detected low level binding of p50 to COX-2 at position −0.25 kb in control macrophages before and after LPS induction (***Figure 6D***). This is in agreement with previous studies demonstrating that

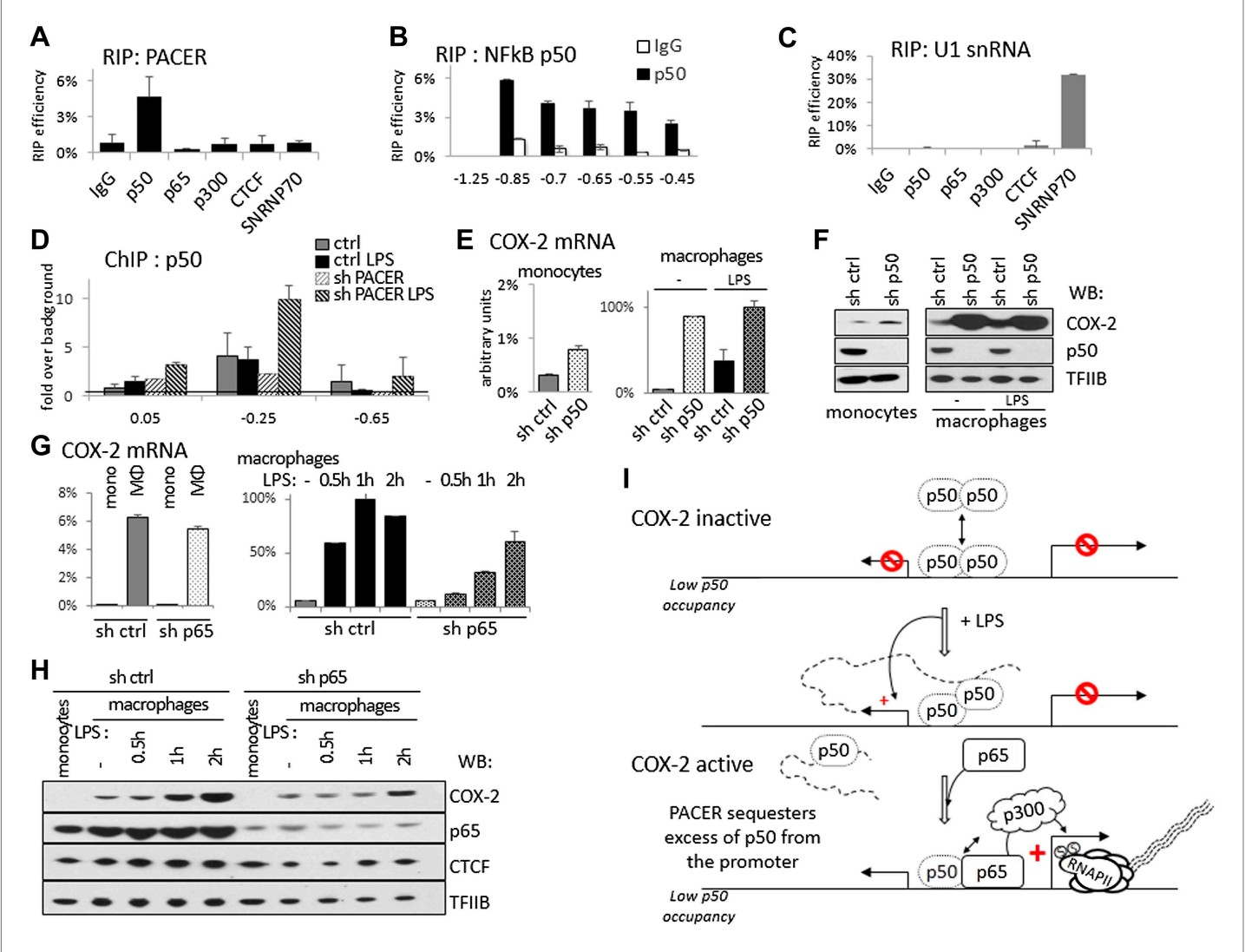

**Figure 6**. PACER controls *COX-2* expression through binding of the repressive NF-κB subunit p50. (**A**) Direct association of p50, p65/RelA, p300, and CTCF with PACER analyzed by RNA immunoprecipitation (RIP) in U937 macrophages. Primers for amplicon −0.85 kb were used. Signal is detected only in p50 immunoprecipitates. (**B**) Direct association of p50 with PACER analyzed by RNA immunoprecipitation (RIP) in U937 macrophages. Sites −0.45 kb through −0.85 kb lie within the lncRNA, while site −1.25 kb is outside and shows no detectable signal. IgG immunoprecipitation is shown as a control. (**C**) Control RIP experiments using the same antibodies as in (**A**) to analyze association with U1 snRNA. (**D**) Association of a small NF-κB subunit p50 with the COX-2 control region was analyzed by ChIP-qPCR in control and sh-lncRNA U937 macrophages. Binding is evident at the −0.25 kb site. (**E** and **F**) Expression of COX-2 mRNA (**E**) and protein (**F**) analyzed in control U937 monocytes and in mock and LPS-stimulated U937 macrophages and upon p50 knockdown. (**G** and **H**) Expression of COX-2 mRNA (**G**) and protein (**H**) analyzed in control U937 monocytes and in mock and LPS-stimulated U937 macrophages and upon p65 knockdown (**I**). A model of PACER-controlled COX-2 activation involving restricting promoter interaction of repressive p50 to facilitate recruitment of p300 HAT and RNAP II pre-initiation complexes to activate mRNA transcription.

p50 levels do not change in response to LPS (*Saccani et al., 2003*; *Saccani et al., 2006*). In untreated PACER-knockdown macrophages, p50 is found at the −0.25 kb site in the COX-2 promoter. However upon LPS stimulation, which normally induces PACER expression, levels of bound p50 increase dramatically when PACER is decreased (*Figure 6D*). This indicates that PACER ensures low p50 association with the COX-2 promoter during activation of the gene.

Why is it necessary to keep binding of p50 'in check' during NF-κB-mediated activation of *COX-2* expression? p50, unlike its dimerization partner p65, lacks an activation domain, and it has been demonstrated that p50 homodimers act as repressors (*Saccani et al., 2006*; *Kastenbauer and Ziegler-Heitbrock, 1999*). We therefore verified whether p50 might repress *COX-2* transcription in the

absence of stimuli that normally induce COX-2 expression. To do this, we established p50-knockdown U937 monocyte lines using lentiviral delivery and measured COX-2 levels in KD cells compared to the control U937. Indeed, knockdown of p50 led to a marked increase of COX-2 mRNA and protein expression (*Figure 6E,F*). As expected, the increase is particularly prominent in undifferentiated U937 monocytes and unstimulated monocytes, whereas upon LPS stimulation COX-2 levels are similar in wt and p50 KD macrophages. These results demonstrate that the small NF-κB subunit p50 by itself functions as a repressor of *COX-2* 'basal', non-induced transcription. To demonstrate the distinct role of p65 subunit on COX-2 expression, we performed lentiviral-mediated knockdown of p65 in U937 cells and measured COX-2 expression in U937 monocytes, macrophages, and during LPS induction (*Figure 6G,H*). Expectedly, p65 knockdown had no influence on 'basal' expression of COX-2 mRNA or protein in monocytes or non-induced macrophages (*Figure 6G*, left and 6H, lanes 1,2 and 6,7). Upon LPS stimulation however, COX-2 induction was severely impaired in p65 knock-down macrophages (*Figure 6G*, right, and 6H lanes 3–5 and 8–10). This indicates that the activating role of p65 is evident only during LPS induction, subsequent to the p50-mediated PACER effects at the COX-2 promoter.

Altogether, these results lead us to postulate a model in which the role of PACER lncRNA in *COX-2* expression is to physically interact and restrict excess p50 from binding the promoter to facilitate the exchange of repressive p50/50 homodimers for p50/p65 heterodimers, thus allowing recruitment of p300, induction of histone hyperacetylation/chromatin remodeling, and consequent assembly of RNAP II complexes competent for transcription activation (*Figure 6G*).

## Discussion

In recent years, there has been an explosion of interest in non-coding RNAs, resulting in the description of distinct ncRNA classes based on their length and function (*Esteller, 2011*). While the activities of small ncRNA species have been well-characterized, the roles of long ncRNAs in gene regulation have been deciphered in only several cases. The predominant model proposes that lncRNAs serve to guide enzymatic complexes to carry out chromatin modifications at specific genes, such as HOTAIR that recruits PRC2 and LSD1 complexes to silence *HOXD* genes and other distant loci (*Tsai et al., 2010*); HOTTIP that recruits MLL1 to activate transcription of *HOXA* genes (*Wang et al., 2011*); and Evf2 ncRNA that recruits the transcription factor DLX2 to activate expression of the *Dlx6* gene (*Bond et al., 2009*). Even in the vicinity of the *COX-2* locus, a recent study identified a species of lncRNA that interacts with the RNA binding proteins hnRNP A/B and hnRNP A2/B1, which might act as transcriptional regulators for a variety of immune-related genes but, interestingly, not *COX-2* (*Carpenter et al., 2013*).

In our study, we describe an unexpected mechanism of gene regulation by an lncRNA (PACER) involving occlusion of a repressor complex (NF-κB p50) from promoter association which then promotes assembly of co-activators (p300 HAT) and RNAP II pre-initiation complexes. PACER directly interacts with p50, which can form both active heterodimers with p65/RelA during the normal course of NF-κB pathway activation, and inactive p50/p50 homodimers which lack transcription activation domains present in p65/RelA. The repressor role of p50 at *COX-2* is consistent with previous studies showing that p50 homodimers inhibit LPS-induced activation of the *TNFα* gene in macrophages and other cells (*Saccani et al., 2006*; *Kastenbauer and Ziegler-Heitbrock, 1999*). Moreover, tumor-associated macrophages, which are refractory to LPS exhibit strong overexpression of p50 and fail to induce TNFα expression despite normal levels of p65/RelA and its ability to re-localize to the nucleus upon LPS treatment (*Saccani et al., 2006*). Importantly, the molecular mechanism by which PACER regulates COX-2 expression does not directly involve the p65/RelA because PACER does not physically interact with this subunit, and the role of p65/RelA is only evident in the induction stage of LPS-mediated COX-2 expression, while PACER mainly affects 'basal', unstimulated COX-2 transcription.

We also describe an additional unexpected mechanism of gene regulation by an lncRNA that involves recruitment of the histone acetyltransferase p300 to catalyze the observed increase in histone acetylation at the *COX-2* locus upon stimulus-induction. We postulate that a large class of activation-related lncRNAs, such as enhancer-associated RNAs (eRNAs), might employ this mode of gene regulation, utilizing NF-κB or other transcription factors as baits to recruit CBP/p300. In this regard, it is interesting to note that enhancers are often co-inhabited by CBP/p300 and lncRNAs (*Kim et al., 2010*; *Visel et al., 2009*). Moreover, CBP/p300 binding, extragenic non-coding transcription, H3K4me1 and increased H3K27 acetylation constitute an active enhancer signature (*Visel et al., 2009*; *Rada-Iglesias et al., 2011*).

Ultimately, the function of the PACER lncRNA is to facilitate assembly of RNA Polymerase II pre-initiation complexes. Upon lncRNA knockdown, the levels of elongating RNAP II and initiating S5P-modified RNAP II are significantly lower, thus affecting the ability of the transcription machinery to efficiently initiate transcription of the *COX-2* gene. While these effects are indirect, they nevertheless demonstrate that PACER belongs to a novel, unanticipated class of *bona-fide* transcriptional regulators.

Our study also describes an unexpected mode of gene regulation by CTCF. CTCF has been previously demonstrated to regulate expression of certain non-coding RNAs (*Sopher et al., 2011*; *Spencer et al., 2011*), but the mechanisms it utilizes remain largely unknown. In this study, we show that CTCF generates a chromatin domain that encompasses the COX-2 core promoter and the upstream regulatory region to restrict adjacent repressive heterochromatin outside of the CTCF-bracketed region. This arrangement ensures induction of the regulatory lncRNA PACER, which orchestrates recruitment of the transcriptional activator p300 and transcription-competent RNAP II complexes. This represents an unanticipated link between CTCF-regulated non-coding RNA and the assembly of RNAP II pre-initiation complexes. Notably, CTCF has been shown to interact with RNAP II and regulate its function, but at the level of transcriptional pausing and alternative splicing (*Chernukhin et al., 2007*; *Wada et al., 2009*), not transcription initiation.

The locations of the two chromatin-bound CTCF/cohesin complexes in the *COX-2* locus suggest that they might perform distinct roles in gene regulation. The −2.5 kb site alone could function in a manner similar to CTCF recognition elements found at the interface between hetero- and euchromatin, such as in the human *p16/CDKN2A, FOXJ3, IGF2/H19, β-GLOBIN, TCR* or *HOXD* loci, where CTCF establishes a chromatin boundary that protects genes from epigenetic silencing (*Witcher and Emerson, 2009*; *Cuddapah et al., 2009*; *Carabana et al., 2011*) or influence from adjacent domains (*Bell and Felsenfeld, 2000*; *Hark et al., 2000*; *Zhong and Krangel, 1999*; *Li and Stamatoyannopoulos, 1994*). On the other hand, the role of CTCF complexes associated with gene promoters is much less understood. Our in vitro analysis of the COX-2 promoter showed that CTCF specifically recognizes a DNA sequence located in the transcribed region, at position +65 (*Figure 2—figure supplements 2, 3*). Similar doublets of CTCF sites, with one located several kilobases upstream and the other within the transcribed region, are found in many genes (data not shown), which could suggest a specific mode of gene regulation by CTCF. We speculate that such doublets participate in establishing a 'promoter-proximal' open chromatin domain, possibly through long range chromatin tethering/looping mediated by two CTCF/cohesin complexes bound to distal sites. Both the enhancer and promoter elements would be positioned inside one chromatin loop, effectively insulated by CTCF/cohesin-mediated boundaries from the surrounding environment, which could restrict the action of enhancers to the appropriate promoter.

Deciphering the mechanisms of *COX-2* gene regulation is of great clinical interest. COX-2 activity has been shown to be instrumental in the development of several types of cancer, including colon, breast, and prostate (*Yap et al., 2010*). For this reason, targeting COX-2 activity with non-steroidal anti-inflammatory drugs (NSAIDs) or agents designed to specifically block COX-2 activity have been approved for therapeutic use (*Salinas et al., 2007*). However, because these drugs have significant off-target side effects, other strategies that specifically modulate *COX-2* transcription could be an attractive alternative approach. For example, *COX-2* downregulation in early stages of tumorigenesis by interfering with PACER-induced COX-2 expression would be predicted to have therapeutic benefits. In addition, *COX-2* silencing has been observed in numerous analyses of tumor specimens and cancer cell lines (*Murata et al., 2004*; *Toyota et al., 2000*; *Meng et al., 2011*). It is attractive to speculate that re-activating *COX-2* expression in advanced stages of cancer by modulating PACER expression would increase COX-2-mediated local tissue mobilization and inflammation, consequently leading to more efficient tumor clearing by immune cells.

## Materials and methods

### Cell culture

Early passages of human mammary epithelial cells were obtained from Martha Stampfer. Cells were grown in MEGM BulletKit (Lonza, Walkersville, MD) media at 37°C in 10% $O_2$ and 5% $CO_2$ and maintained according to procedures outlined on the Stampfer laboratory website (http://HMEC.lbl.gov/other/procs.html). Human histiocytic lymphoma monocyte cell line U937 (*Sundstrom and Nilsson,*

*1976*) was kindly provided by Dr Sebastien Landry. U937 cells were maintained in RPMI medium (Mediatech, Corning, Manassas, VA) supplemented with 10% fetal bovine serum and antibiotics. For macrophage differentiation, cells were seeded onto plates at $0.5 \times 10^6$/ml and treated with 0.2 mM phorbol myristate acetate (PMA, Sigma, St. Louis, MO) for 48 hr, followed by 24 hr incubation without PMA. To induce COX-2 expression, U937-derived macrophages were treated with LPS (2 µg/ml) for 6 hr.

## ChIP

Cells in exponential growth phase (HMEC, undifferentiated U937) or differentiated (macrophages) were cross-linked with 1% formaldehyde in 5 mM Hepes pH 8.0, 10 mM NaCl, 0.1 mM EDTA, 50 µM EGTA for 10 min at RT. Cross-linking was stopped with 0.125M glycine. Adherent cells were washed two times with PBS and collected by scraping into ice-cold PBS followed by centrifugation at 300×*g* for 5 min. Suspension cells (U937) were washed two times and collected by centrifugation at 300×*g* for 5 min at 4°C. Cells were lysed by rotating for 10 min on ice in 50 mM K-Hepes pH 7.5, 140 mM NaCl, 1 mM EDTA, 10% glycerol, 0.5% NP-40, 0.25% Triton X-100, and protease inhibitors and then centrifuged at 600×*g* for 10 min at 4°C. Chromatin pellets were washed once in 10 mM Tris–HCl pH 8.0, 0.2M NaCl, 0.1 mM EDTA, 50 µM EGTA and resuspended in 10 mM Tris–HCl pH 8.0, 0.1% NP-40, 0.1 mM EDTA, 50 µM EGTA. Sonication (20 rounds of 10s at power setting 3.5) was performed with a Branson sonicator, followed by centrifugation to clear the chromatin solution from debris. An aliquot of chromatin corresponding to $10^6$ cells was brought up to 1 ml with 20 mM HEPES, pH7.9, 0.2M NaCl, 2 mM EDTA, 0.1% Na-DOC, 0.5% Triton X-100, 1 mg/ml BSA, and protease inhibitors and incubated with 2 µg of antibodies overnight at 4°C. To capture immune complexes, 10 µl of protein G beads was added and incubated for 2 hr at 4°C. Beads were washed (wash 1: 20 mM HEPES, pH7.9, 0.2M NaCl, 2 mM EDTA, 0.1% Na-DOC, 0.5% Triton X-100; wash 2: same as wash 1 but with 0.4M NaCl; wash 3: 20 mM Tris–HCl, pH 8.0, 0.25M LiCl, 2 mM EDTA, 0.25% Na-DOC; wash 4: TE, 0.1% NP40), immune complexes were detached from beads by 10 min incubation at 65°C in 0.1M Tris–HCl, pH 8.0, 1% SDS, followed by adding 140 µg of Proteinase K (NEB, Ipswich, MA) and NaCl to 0.1M, and incubation for 2 hr at 42°C and then overnight at 65°C. DNA was purified by phenol/chloroform extraction and analyzed by qPCR. IP efficiencies were calculated as a ratio between qPCR signals obtained with IP material and input DNA processed in parallel with ChIP samples. The following antibodies were used in ChIP experiments: CTCF (Millipore, Billerica, MA; Cell Signalling, Danvers, MA; Santa Cruz Biotechnology, Dallas, TX), p50 (Abcam, Cambridge, MA), Rad21 (Abcam), RNAP II, H2AZ (Active Motif, Carlsbad, CA), H3K4me2, H3K4me3, H3K9me3, H4K20me3, H4K8Ac, H3Ac (Abcam), p300-Ac (Cell Signaling). qPCR primers are listed in *Supplementary file 1*. All ChIP experiments were repeated three times and qPCR reactions were performed in triplicates.

## Subcellular fractionation

Cells were scraped into ice-cold PBS, collected by centrifugation at 300×*g* for 5 min at 4°C, resuspended in 5 vol of cytoplasmic extract buffer (CEB, 10 mM HEPES pH 7.9, 50 mM NaCl, 1 mM EDTA, 0.3% NP40, 2.5 mM $MgCl_2$, 1 mM DTT, protease and RNAse inhibitors) and kept on ice for 5 min. After centrifugation at 600×*g* for 5 min at 4°C, the cytoplasmic fraction was collected and the nuclear pellet was resuspended in nuclear extract buffer (NEB, 10 mM HEPES pH 7.9, 500 mM NaCl, 1 mM EDTA, 1 mM EGTA, 0.1% NP40, 0.5% Triton X-100, 2.5 mM $MgCl_2$, 1 mM DTT, protease and RNAse inhibitors), kept on ice for 10 min and centrifuged for 10 min at max speed. The nucleoplasmic fraction was collected and the pellet was resuspended in the same buffer, sonicated five times for 10 s at power 3 and cleared by centrifugation to produce the chromatin-bound fraction.

## mRNA and lncRNA quantification

RNA purification was performed with Trizol (Life Technologies, Carlsbad, CA) according to the manufacturer's protocol. cDNA was produced using the Superscript III system (Life Technologies) using random hexamers or strand-specific primers listed in *Supplementary file 1*. Relative quantities of mRNA were calculated using standard curves generated with cDNAs containing the highest concentration of a given target. For the ncRNA, the standard curves were generated with genomic DNA. The results are presented as the mean ± s.d of three independent RNA preparations.

## ChIP-chip

ChIP DNA and reference input DNA were amplified by ligation-mediated PCR according to NimbleGen (Roche, Madison, WI) recommendations. Briefly, 1 µg of DNA was blunted with T4 DNA polymerase (NEB),

double-stranded staggered oligonucleotide adaptors were ligated, and PCR amplification was performed with oligos specific to the adaptors using a mixture of Taq polymerase (NEB) and PfuTurbo (Stratagene, Agilent Technologies, Santa Clara, CA ). 4 µg of each DNA was purified and sent to NimbleGen for probe preparation and hybridization. LM-PCR amplified DNA was verified to be enriched for target sequences that were detected by ChIP.

Custom high density genomic tiling arrays contained ~80 regions covering selected genes and 50 or 100 kb flanks on either side. Array manufacturing, probe labeling, hybridization, and signal acquisition were conducted by NimbleGen. Data were visualized using SignalMap software (NimbleGen, Roche).

### Reporter gene assays
Constructs were cloned into the pGL3basic vector (Promega, Madison, WI). Plasmids were transfected in a 20:1 ratio (pGL3 to control Renilla Luc pRL-TK vector) using Lipofectamine 2000 (Life Technologies) into HEK293T cells in 96-well dishes and assayed 24 hr post transfection using the Dual-Glo Luciferase assay system (Promega). All transfections were done in five replicates.

### Western blotting
Western blotting was performed using standard protocols using the following antibodies: CTCF (Becton Dickinson, Franklin Lakes, NJ), COX-2 (Cayman, Ann Arbor, MI), p50 (Abcam), TFIIB (Santa Cruz), actin (Sigma).

### CTCF binding site prediction
Computer-based site prediction was performed using the online tool at http://insulatordb.uthsc.edu/.

### RACE
5′ and 3′ ends of COX-2 lncRNA were mapped using Rapid Amplification of cDNA ends (RACE) kit (Roche) according to the manufacturer's recommendations. Sequences of primers used for strand-specific cDNA synthesis and RACE amplification reactions are listed in *Supplementary file 1*. The resulting single RACE products (from five independent 5′ and three independent 3′ amplifications) were gel-purified and sequenced.

### RNA immunoprecipitation (RIP)
RIP experiments were performed using MagnaRIP RNA-Binding Protein Immunoprecipitation Kit (Millipore) according to the manufacturer's recommendations. Antibodies were purchased from Santa Cruz Biotechnology (p50, p65, p300) and Millipore (CTCF). qPCR primers are listed in *Supplementary file 1*. All experiments were repeated three times and qPCR reactions were performed in triplicates.

### siRNA
siRNA against CTCF, custom-designed siRNA against PACER and control-scrambled siRNAs were purchased from Dharmacon (Thermo Fisher, Waltham, MA). siRNAs against SMC3 were purchased from Ambion (Life Technologies). Sequences of custom siRNA are listed in *Supplementary file 1*. $1 \times 10^5$/ml HMECs were transfected according to the manufacturer's protocol using DharmaFect I reagent. For PACER, four siRNA were designed and tested, from which two efficiently knocked down lncRNA expression (siPACER-753 and siPACER-870, data not shown). The siPACER-753 was most efficient and was used in all experiments. Cells were harvested 72 hr after transfection.

### shRNA
Short hairpin sequence against COX-2 lncRNA was created by linking sense siRNA 870, loop, antisense siRNA870 and RNAP II terminator sequences; and cloning into BamHI and EcoRI sites in pGreenPuro lentiviral vector (SBI, Mountain View, CA) downstream of the H1 promoter. shRNA against the Luciferase gene was used as a control. shRNAs against p50 and p65 were purchased from Sigma. U937 monocytes were transduced with lentiviral particles and selected with puromycin. Once stable lines were established, puromycin was removed at least 48 hr before harvesting or differentiation.

## Acknowledgements
We gratefully acknowledge Dr Martha Stampfer for providing variant HMECs. This work was supported by grants from NIH (R01 CA159354), the Chambers Medical Foundation, and the GemCon Foundation to BME.

# Additional information

## Funding

| Funder | Grant reference number | Author |
| --- | --- | --- |
| National Institutes of Health | R01 CA159354 | Beverly M Emerson |
| Chambers Medical Foundation | | Beverly M Emerson |
| GemCon Foundation | | Beverly M Emerson |

The funder had no role in study design, data collection and interpretation, or the decision to submit the work for publication.

## Author contributions

MK, Conception and design, Acquisition of data, Analysis and interpretation of data, Drafting or revising the article; BME, Conception and design, Analysis and interpretation of data, Drafting or revising the article

# Additional files

## Supplementary file

• Supplementary file 1. Sequences of oligonucleotides used in this study

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
