## [Decision Letter]

Thank you for sending your work entitled “p50-Associated COX2 Extragenic RNA (PACER) activates *COX2* gene expression by sequestering repressive NFκB complexes” for consideration at *eLife*. Your article has been favorably evaluated by a Senior editor and 3 reviewers, one of whom is a member of our Board of Reviewing Editors.

The Reviewing editor and the other reviewers discussed their comments before we reached this decision, and the Reviewing editor has assembled the following comments to help you prepare a revised submission:

The three reviewers agreed that the paper is exciting and interesting, as it describes a novel lncRNA involved in control of a gene locus of significant biomedical relevance by an unprecedented mechanism. The manuscript reports the characterization of the lncRNA dubbed PACER, which originates from the upstream region of the COX-2 gene in the antisense direction. It is demonstrated that PACER positively regulates COX-2 expression. Mechanistically, PACER promotes histone modifications associated with gene activation and RNAPII association with the COX2 promoter region. PACER binds to the DNA binding protein p50, and it is proposed that this binding leads to gene activation by favoring formation of active p50-p65 NFKB heterodimers, which in turn would drive COX2 transcription. The reviewers agreed that these results merit publication in a high-impact journal such as *eLife*, but they also concurred in that additional work is required to strengthen the main conclusions of this interesting paper.

The reviewers agreed to summarize their request for new work as follows:

1) A better characterization of PACER is needed. Key questions raised by the reviewers are: What is the size, exact sequence, relative abundance and exact intracellular distribution of PACER? Does PACER originate from its own unique promoter, is it a divergent transcript from the COX-2 promoter, or is it an eRNA? To address these issues, the reviewers propose the following work:

1.1) Full sequence data from the RACE experiments should be provided.

1.2) A Northern blot analysis and/or qPCR analysis should be performed to determine the abundance of PACER relative to COX-2 mRNA.

1.3) A better characterization of the PACER transcription start site is needed. ChIP analysis of RNAPII and chromatin modifications are provided in the manuscript, but these assays are correlative in nature and could not fully elucidate whether PACER originates from its own unique promoter, if it is a divergent transcript from the COX-2 promoter, or if it is an eRNA. This issue can be resolved with promoter reporter assays to define the DNA sequence that drives PACER transcription.

1.4) Both the RNA and RNAPII ChIP profile suggest a longer transcript than what was cloned by RACE, up to -2 kb. The authors should explain how this matches the suggested 3' end of the RNA at -1022 and show the RACE data rather than just describing it. Is there a canonical cleavage and polyadenylation signal around -1022?

1.5) Strand-specific amplification of the plus strand should include an oligo within the COX-2 gene as a positive control.

1.6) Figure 4 shows the subcellular localization of PACER and COX-2 as determined by cellular fractionation. The authors should perform controls of transcripts present only in the nucleus and only in the cytoplasm to account for cross-contamination of the fractions.

1.7) The reviewers wonder whether PACER has been detected by recent genome wide Global Run-On deep sequencing experiments (GRO-seq) done in epithelial cells. Publicly available GRO-seq datasets that could be mined to answer this question include Galbraith MD et al, Cell. 2013 Jun 6;153(6):1327-39 (performed in HCT116 colon cancer cells) and the various papers from the Kraus team using MCF7 breast cancer cells such as Danko CG Mol Cell. 2013 Apr 25;50(2):212-22. GRO-seq results for the COX-2 (PTGS2) locus could be displayed to further illuminate the site of PACER transcription.

2) The model of PACER action needs further experimental validation.

2.1) Whereas the repressive role of p50 is well demonstrated, the positive role of p65 NFKB in driving COX2 expression is not demonstrated in the context of PACER action. As currently depicted in Figure 6, it is p65 that recruits p300 to drive histone modifications and COX2 transcription. The reviewers request that the role of p65 on PACER action be demonstrated via knockdown.

2.2) Additionally, how does PACER knockdown or CTCF knockdown affect p65 binding to the COX2 promoter? Reviewers propose that ChIP assays for p65 be performed to answer this question.

3) A better presentation and interpretation of the ChIP data is needed.

3.1) CTCF ChIPs. Regarding the ChIP-on-chip experiment, the authors should show at least one negative control and one positive control locus out of the “numerous other genes” on the chip and extend the COX-2 analysis several kb up and downstream to demonstrate the specificity of the IP. There is a discrepancy between the ChIP-on-chip and ChIP-qPCR analysis for CTCF binding sites. In the ChIP-on-chip, one sees a clear enrichment of CTCF binding at least to -500bp, which is not confirmed by the qPCR. Also, the peak for CTCF binding around TSS is much more pronounced than the one around -2.5kb. Again, this is not recapitulated in the ChIP-qPCR. In addition, the claim that Rad21 binds around the TSS along with CTCF is not substantiated by the ChIP presented in Figure 2, especially if one takes into account the IgG background that differs between different amplicons. The authors should aim to explain the possible origin of these discrepancies.

3.2) Do the two described peaks overlap with a CTCF consensus motif?

3.3) It is stated in the text that H4K20me3 is reduced upon CTCF knockdown; Figure 3, however, shows an increase in H4K20me3. The text reads: “This modification was reduced at both sites upon CTCF knockdown...”, whereas the data shows a clear increase of H4K20me3, although only at the distal peak at -2.5 kb. Please revise the description of this result.

3.4) Chromatin modifications are evaluated upon PACER knockdown. Although the manuscript claims “no significant changes in H3K4me2”, there are clear differences in Figure S5 (not Figure S4 as stated in the text), especially at the important promoter and -2.5 kb sites. Please revise the description of this result.

3.5) The conclusion of PACER directly recruiting RNAP II is not supported by the data shown. Instead, the changes in chromatin modification might indirectly influence RNAP II association. The text should be revised to acknowledge this.

---

## [Author Response]

*1) A better characterization of PACER is needed. Key questions raised by the reviewers are: What is the size, exact sequence, relative abundance and exact intracellular distribution of PACER? Does PACER originate from its own unique promoter, is it a divergent transcript from the COX-2 promoter, or is it an eRNA? To address these issues, the reviewers*
*propose the following work:*

*1.1) Full sequence data from the RACE experiments should be provided*.

The full sequence of PACER, as well as sequencing data (raw sequence alignments and sequencing traces) for several independent 5’ and 3’ RACE reactions are now provided in Figure 1—figure supplement 1, Figure 1—figure supplement 2 and Figure 1—figure supplement 3.

*1.2) A Northern blot analysis and/or qPCR analysis should be performed to determine the abundance of PACER relative to COX-2 mRNA*.

We used qPCR to accurately measure relative amounts of PACER and COX-2 mRNA in HMEC and U937 macrophages. The results for HMEC are depicted in Figure 1. The abundance of PACER RNA is roughly 30 fold lower than that of mRNA, both in HMEC and macrophages. Of note, we also attempted to use Northern blotting to detect PACER expression, however those experiments were inconclusive, as the signal for PACER was at or below the level of detection. We believe this is because PACER exists as a mixture of molecules differing at their 5’ ends, as shown by RACE mapping. Because such mixtures migrate as a ‘smear’ in gels, they are difficult to detect, especially if present at very low abundance.

*1.3) A better characterization of the PACER transcription start site is needed. ChIP analysis of RNAPII and chromatin modifications are provided in the manuscript, but these assays are correlative in nature and could not fully elucidate whether PACER originates from its own unique promoter, if it is a divergent transcript from the COX-2 promoter, or if it is an eRNA. This issue can be resolved with promoter reporter assays to define the DNA sequence that drives PACER transcription*.

We now provide additional pieces of evidence and explanation demonstrating that PACER is a nuclear lncRNA synthesized from its own RNAPII dependent promoter. We performed reporter gene assays which indicate that a separate promoter located ∼0.3-0.5kb can drive expression of PACER independently of the COX-2 mRNA promoter (Figure 1). In addition, RNAPII ChIP assays have been re-formatted to allow easier visualization of a separate peak of S5P-modified RNAP II at position ∼-0.4kb (Figures 1 and 5).

*1.4) Both the RNA and RNAPII ChIP profile suggest a longer transcript than what was cloned by RACE, up to -2 kb. The authors should explain how this matches the suggested 3' end of the RNA at -1022 and show the RACE data rather than just*
*describing it. Is there a canonical cleavage and polyadenylation signal around -1022?*

Indeed, the COX-2 upstream region is transcriptionally active extending to around -2kb, as suggested by our RT-qPCR analyses, RNAPII ChIP and available GRO-SEQ data. However, we believe that PACER is the most abundant well-defined message based on the GRO-SEQ results and considering that we failed to clone additional transcripts further upstream using RACE (data not shown). PACER sequence contains a well conserved polyadenylation signal at position -999 (Figure 1—figure supplement 1) and a well-defined 3’ end at -1022. Nevertheless, it is possible that additional transcripts exist further upstream. These are unlikely to function in the same manner, because we did not recover RNAs mapping to that area in p50 immunoprecipitates (e.g., Figure 6, and data not shown). These issues are now also discussed in the Results section under Figure 1.

*1.5) Strand-specific amplification of the plus strand should include an oligo within the COX-2 gene as a positive control*.

This control is now included as Figure 1.

*1.6)*
Figure 4
*shows the subcellular localization of PACER and COX-2 as determined by cellular fractionation. The authors should perform controls of transcripts present only in the nucleus and only in the cytoplasm to account for cross-contamination of the fractions*.

Figure 4 now includes control amplification of nuclear (COX-2 intron) and cytoplasmic RNA (mRNA for actin). These results clearly indicate that PACER is predominantly nuclear, as its profile closely resembles that of the intronic RNA. Cytoplasmic RNAs are found in the nuclear fractions to some extent, likely because a fraction of mature RNA remains within the spliceosome/nuclear export complexes and co-purifies with the nuclear fractions, this is clearly visible for COX-2 mRNA but also actin mRNA. Conversely, small amounts of nuclear RNAs are found in the cytoplasmic fraction. This is likely to occur during biochemical separation of fractions, where small molecules are known to passively leak out through nuclear pores. Despite this minor cross-contamination, which is evident for a nucleus-exclusive intronic RNA, we believe our data supports the conclusion that PACER is localized to the nucleus. This discussion of this issue is included in the text under Figure 4. In addition, recent experiments from our laboratory indicate that PACER is not associated with cytoplasmic polysomal fractions or with mono-ribosomes (D. Henderson, personal communication), further indicating that it is confined to the nucleus. This point is mentioned in the Discussion.

*1.7) The reviewers wonder whether PACER has been detected by recent genome wide Global Run-On deep sequencing experiments (GRO-seq) done in epithelial cells. Publicly available GRO-seq datasets that could be mined to answer this question include Galbraith MD et al, Cell. 2013 Jun 6;153(6):1327-39 (performed in HCT116 colon cancer cells) and the various papers from the Kraus team using MCF7 breast cancer cells such as Danko CG Mol Cell. 2013 Apr 25;50(2):212-22. GRO-seq results for the COX-2 (PTGS2) locus could be displayed to further* illuminate the site of PACER transcription.

Antisense transcription upstream of the COX-2 transcription start site was indeed detected in HCT116 cells in the recent study [Galbraith, M.D., et al., *HIF1A employs CDK8-mediator to stimulate RNAPII elongation in response to hypoxia.* Cell, 2013. 153(6): p. 1327-39.]. These signals extend up ∼1100bp upstream, therefore perfectly correlating with the presence of PACER in this region. A genome browser picture incorporating these data is now included as Figure 1—figure supplement 4 and it is referenced in the main text under Figure 1.

*2) The model of PACER action needs further experimental validation*.

*2.1) Whereas the repressive role of p50 is well demonstrated, the positive role of p65 NFKB in driving COX2 expression is not demonstrated in the context of PACER action. As currently depicted in*
Figure 6*, it is p65 that recruits p300 to drive histone modifications and COX2 transcription. The reviewers request that the role of p65 on PACER action be demonstrated via knockdown*.

As requested by the reviewers, we performed p65 knock-down experiments in U937 monocytes and analyzed its effects on COX-2 expression in the context of PACER function (Figure 6). These results clearly indicate that the role of p65 at the COX-2 promoter is confined to the induction phase, because in non-induced macrophages the knockdown had no effects upon COX-2 expression, contrary to what we observed for p50 (Figure 6 vs 6E). This is to be expected because, while p50 remains bound to the COX-2 promoter in non-induced conditions, p65 is retained in the cytoplasm by IκB complexes [Gilmore, T.D., *Introduction to NF-kappaB: players, pathways, perspectives.* Oncogene, 2006. 25(51): p. 6680-4]. Only upon activation of the NF-κB signaling pathway p65 relocates into the nucleus and activates its target genes [Gilmore, T.D., *Introduction to NF-kappaB: players, pathways, perspectives.* Oncogene, 2006. 25(51): p. 6680-4]. Contrary to this, the activation role of PACER on COX-2 expression is evident both in non-induced macrophages (Figure 4) and in HMECs (Figure 4). In addition, RIP experiments suggest that PACER does not contact p65 (Figure 6). We therefore believe that the role of PACER is independent of p65; as PACER occludes p50, effectively inactivating it as a repressor, and allows activators (p65 in the case of LPS stimulation in macrophages, and possibly other transcription factors in the case of HMECs) to induce RNAP II recruitment and mRNA transcription.

*2.2) Additionally, how does PACER knockdown or CTCF knockdown affect p65 binding to the COX2 promoter? Reviewers propose that ChIP assays for p65 be performed to answer this question*.

Despite repeated efforts we were not able to demonstrate binding of p65 to the COX-2 locus. We performed these ChIP experiments before the original submission of our manuscript with no success, and repeated again in response to reviewers’ comments. We are well aware that other studies have been performed where ChIP was used to demonstrate p65 binding, including genome-wide studies in mice and humans [Xing, Y., et al., *Characterization of genome-wide binding of NF-kappaB in TNFalpha-stimulated HeLa cells.* Gene, 2013. 526(2): p. 142-9; Lim, C.A., et al., *Genome-wide mapping of RELA(p65) binding identifies E2F1 as a transcriptional activator recruited by NF-kappaB upon TLR4 activation.* Mol Cell, 2007. 27(4): p. 622-35; Schreiber, J., et al., *Coordinated binding of NF-kappaB family members in the response of human cells to lipopolysaccharide.* Proc Natl Acad Sci U S A, 2006. 103(15): p. 5899-904; Kaikkonen, M.U., et al., *Remodeling of the enhancer landscape during macrophage activation is coupled to enhancer transcription.* Mol Cell, 2013. 51(3): p. 310-25]. We attempted to perform similar analyses using the antibodies that were used by other investigators, including, but not limited to, sc-372X from Santa Cruz or Ab7970 from Abcam, all without success. We consistently observed background signals at COX-2, as well as at other regions where p65 binding was also expected, including numerous “positive control” regions that exhibited particularly strong signals for p65 in previous studies (TNFa, IL-2, CCL5 and CSF5). Since our laboratory possesses extensive expertise in ChIP and these types of experiments are performed routinely with success, we were forced to conclude that currently available antibody lots do not permit us to perform p65 ChIP experiments. However, given that the role of p65 in the context of COX-2 expression has been studied extensively [Schmedtje, J.F., Jr., et al., *Hypoxia induces cyclooxygenase-2 via the NF-kappaB p65 transcription factor in human vascular endothelial cells.* J Biol Chem, 1997. 272(1): p. 601-8; D'Acquisto, F., et al., *Involvement of NF-kappaB in the regulation of cyclooxygenase-2 protein expression in LPS-stimulated J774 macrophages.* FEBS Lett, 1997. 418(1-2): p. 175-8; Shishodia, S. and B.B. Aggarwal, *Cyclooxygenase (COX)-2 inhibitor celecoxib abrogates activation of cigarette smoke-induced nuclear factor (NF)-kappaB by suppressing activation of IkappaBalpha kinase in human non-small cell lung carcinoma: correlation with suppression of cyclin D1, COX-2, and matrix metalloproteinase-9.* Cancer Res, 2004. 64(14): p. 5004-12; Jang, B.C., et al., *Induction of cyclooxygenase-2 in macrophages by catalase: role of NF-kappaB and PI3K signaling pathways.* Biochem Biophys Res Commun, 2004. 316(2): p. 398-406] and our knock-down experiments clearly indicate a p65 role in the induction of COX-2 expression, we believe that our experiments provide sufficient support for the model of PACER action.

*3) A better presentation and interpretation of the ChIP data is needed*.

*3.1) CTCF ChIPs. Regarding the ChIP-on-chip experiment, the authors should show at least one negative control and one positive control locus out of the “numerous other genes” on the chip and extend the COX-2 analysis several kb up and downstream to demonstrate the specificity of the IP. There is a discrepancy between the ChIP-on-chip and ChIP-qPCR analysis for CTCF binding sites. In the ChIP-on-chip, one sees a clear enrichment of CTCF binding at least to -500bp, which is not confirmed by the qPCR. Also, the peak for CTCF binding around TSS is much more pronounced than the one around -2.5kb. Again, this is not recapitulated in the ChIP-qPCR. In addition, the claim that Rad21 binds around the TSS along with CTCF is not substantiated by the ChIP presented in*
Figure 2*, especially if one takes into account the IgG background that differs between different amplicons. The authors should aim to explain the possible origin of these discrepancies*.

The ChIP-on-chip data representing results at three loci, COX-2, c-myc and INSIG2 are now shown in Figure 2—figure supplement 1 Each region on the array was tiled with ∼50bp probes with approximately 5x coverage, apart from repeated DNA, which was not covered. As a result, a 50kb region contains up to 250.000 probes. Each probe that shows background signal effectively serves as a ‘negative control’ for any other probe on the array. Nevertheless, as requested, we show a gene (INSIG2) where no enrichment for CTCF is present within ∼25kb upstream or downstream of its promoter. Given the tiling density, we are confident that signals shown as peaks represent true enrichment of a DNA region in the immunoprecipitates. That said, a caveat might exist that an unusually high GC content of a region (such as a CpG island around COX-2 promoter) will cause stronger hybridization resulting in unusually high signal for relatively lowly enriched DNA. This, in our opinion, might cause the apparent discrepancy between the strength of COX-2 promoter signal on the array as compared to qPCR data. For this reason, we tend to believe that the strong ĆTCF peak extending up to ∼500bp upstream of the COX-2 promoter in ChIP-on-chip data is, in reality, weaker and represents binding of CTCF at the position +65bp, as mapped by ChIP-qPCR walking, EMSA and DNA recruitment assays (Figure 1, Figure 2—figure supplement 2 and Figure 2—figure supplement 3). These considerations are now also included in the text under Figure 2.

Concerning Rad21, we performed ChIP-qPCR walking several times and we have consistently seen Rad21 binding to COX-2 promoter, albeit at the lower level than at the -2.5kb site. This low level is still significantly higher than any other negative site, and any IgG signal. We calculated statistical significance levels comparing signal values at the COX-2 promoter versus two other negative sites at -1.55kb and -0.85kb and found p values equal to 0.028 and 0.033, respectively. These values are now depicted in Figure 2.

*3.2) Do the two described peaks overlap*
*with a CTCF consensus motif?*

We found CTCF binding consensus motifs located both within the -2.5kb peak (GCAGCAGAAGGGGGCAGTA at -2508bp) and at the promoter (GCAGCGCCTCCTTCAGCTCCA at +65). This information is now mentioned in the text under Figure 2.

*3.3) It is stated in the text that H4K20me3 is reduced upon CTCF knockdown;*
Figure 3*, however, shows an increase in H4K20me3. The text reads: “This modification was reduced at both sites upon CTCF knockdown...”, whereas the data shows a clear increase of H4K20me3, although only at the distal peak at -2.5 kb. Please revise the description of this result*.

The text has been revised and now reads: “A repressive mark H4K20me3 had a more localized pattern, with two discernible peaks at -2.5kb and ∼-1.2kb. This modification was increased at -2.5kb upon CTCF knockdown (Figure 3).”

*3.4) Chromatin modifications are evaluated upon PACER knockdown. Although the manuscript claims “no significant changes in H3K4me2”, there are clear differences in Figure S5 (not Figure S4 as stated in the text), especially at the important promoter and -2.5 kb sites. Please revise the description of this result*.

We have now revised the text to mention those changes in H3K4me2. The text now reads “ChIP-qPCR experiments showed no significant changes in H3K4 trimethylation or H2AZ association (data not shown) and a modest effect on H3K4 dimethylation (Figure 5—figure supplement 2).” We have not focused our attention on these findings because in the absence of H3K4me3 changes, we found it difficult to interpret H3K4me2 differences alone. Moreover, H3K4me2 is affected at the promoter in monocytes and macrophages, while at -2.5kb the difference is only evident after LPS induction. Such distinct effects clearly need further investigation, but are very unlikely to participate in PACER-modulated regulation of COX-2 expression.

*3.5) The conclusion of PACER directly recruiting RNAP II is not supported by the data shown. Instead, the changes in chromatin modification might indirectly influence RNAP II association. The text should be revised to acknowledge this*.

We fully agree with the notion that the effects in RNAP II recruitment are in all likelihood indirect, possibly mediated through p300-induced changes in histone acetylation. We have now revised the text to reflect this point more clearly. The text now reads: “To explore the possibility that PACER could affect (directly or indirectly) the assembly of RNA Polymerase II preinitiation complexes and/or later steps in transcriptional activation […] These results demonstrate that PACER influences early steps in transcription initiation at the stage of formation of RNAP II preinitiation complexes, likely through p300-induced changes in promoter-associated histone acetylation.” Also, in the Discussion section “Ultimately, the function of PACER lncRNA is to facilitate assembly of RNA Polymerase II pre-initiation complexes. Upon lncRNA knockdown, the levels of elongating RNAP II and initiating S5P-modified RNAP II are significantly lower, thus affecting the ability of the transcription machinery to efficiently initiate transcription of the COX-2 gene. While these effects are indirect, they nevertheless demonstrate that PACER belongs to a novel, unanticipated class of bona-fide transcriptional regulators.”